# Membership Inference Attacks against Fine-tuned Large Language Models via Self-prompt Calibration

**Wenjie Fu**
Huazhong University of
Science and Technology
wjfu99@outlook.com

**Huandong Wang**[*]
Tsinghua University
wanghuandong@tsinghua.edu.cn

**Chen Gao**
Tsinghua University
chgao96@gmail.com

**Guanghua Liu**
Huazhong University of
Science and Technology
guanghualiu@hust.edu.cn

**Yong Li**
Tsinghua University
liyong07@tsinghua.edu.cn

**Tao Jiang**
Huazhong University of
Science and Technology
taojiang@hust.edu.cn

## Abstract

Membership Inference Attacks (MIA) aim to infer whether a target data record has been utilized for model training or not. Existing MIAs designed for large language models (LLMs) can be bifurcated into two types: reference-free and reference-based attacks. Although reference-based attacks appear promising performance by calibrating the probability measured on the target model with reference models, this illusion of privacy risk heavily depends on a reference dataset that closely resembles the training set. Both two types of attacks are predicated on the hypothesis that training records consistently maintain a higher probability of being sampled. However, this hypothesis heavily relies on the overfitting of target models, which will be mitigated by multiple regularization methods and the generalization of LLMs. Thus, these reasons lead to high false-positive rates of MIAs in practical scenarios. We propose a Membership Inference Attack based on Self-calibrated Probabilistic Variation (SPV-MIA). Specifically, we introduce a self-prompt approach, which constructs the dataset to fine-tune the reference model by prompting the target LLM itself. In this manner, the adversary can collect a dataset with a similar distribution from public APIs. Furthermore, we introduce probabilistic variation, a more reliable membership signal based on LLM memorization rather than overfitting, from which we rediscover the neighbour attack with theoretical grounding. Comprehensive evaluation conducted on three datasets and four exemplary LLMs shows that SPV-MIA raises the AUC of MIAs from 0.7 to a significantly high level of 0.9. Our code and dataset are available at: https://github.com/tsinghua-fib-lab/NeurIPS2024_SPV-MIA.

## 1 Introduction

Large language models (LLMs) have been validated to have the ability to generate extensive, creative, and human-like responses when provided with suitable input prompts. Both commercial LLMs (e.g., ChatGPT [51]) and open-source LLMs (e.g., LLaMA [65]) can easily handle various complex application scenarios, including but not limited to chatbots [20], code generation [66], article co-writing [28]. Moreover, as the pretraining-finetuning paradigm becomes the mainstream pipeline in of LLM field, small-scale organizations and individuals can fine-tune pre-trained models over their private datasets for downstream applications [44], which further enhances the influence of LLMs.

---

[*]Corresponding author.

38th Conference on Neural Information Processing Systems (NeurIPS 2024).

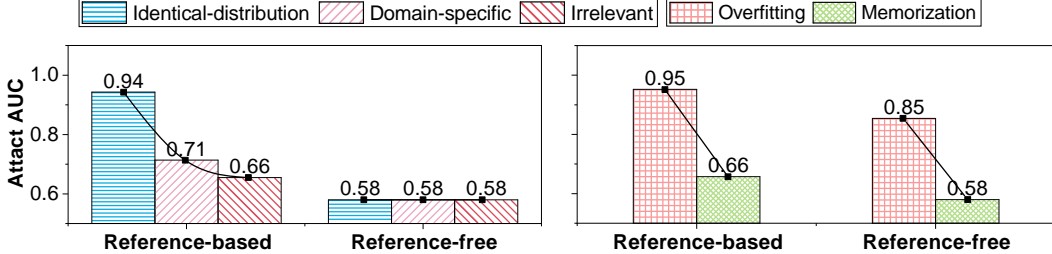

(a) AUC w.r.t reference dataset source, which is utilized to fine-tune reference model for difficulty calibration

(b) AUC w.r.t training phase, where memorization is a stage inevitable and arises before overfitting

Figure 1: Attack performances of the reference-based MIA (LiRA [8, 46, 47, 72]) and reference-free MIA (LOSS Attack [73]) unsatisfy against LLMs in practical scenarios, where LLMs are in the memorization stage and only domain-specific dataset is available. (a) Reference-based MIA shows a catastrophic plummet in performance when the similarity between the reference and training datasets declines. (b) Existing MIAs are unable to pose privacy leakages on LLMs that only exhibit memorization, an inevitable phenomenon occurs much earlier than overfitting and persists throughout almost the entire training phase [47, 64, 75].

However, while we enjoy the revolutionary benefits raised by the popularization of LLMs, we also have to face the potential privacy risks associated with LLMs. Existing work has unveiled that the privacy leakage of LLMs exists in almost all stages of the LLM pipeline [9, 19, 41, 53, 59, 61]. For example, poisoning attacks can be deployed during pre-training, distillation, and fine-tuning [34, 68]. Moreover, data and model extraction attacks can be conducted through inference [6, 22]. Among these attacks, fine-tuning is widely recognized as the stage that is most susceptible to privacy leaks since the relatively small and often private datasets used for this process [74]. Therefore, this paper aims to uncover the underlying privacy concerns associated with fine-tuned LLMs through an exploration of the membership inference attack (MIA).

MIA is an adversary model that categorizes data records into two groups: member records, which are included in the training dataset of the target model, and nonmember records, which belong to a disjoint dataset [60]. MIAs have been well studied in classic machine learning tasks, such as classification, and reveal significant privacy risks [25]. Recently, some contemporaneous works attempt to utilize MIAs to evaluate the privacy risks of LLMs. For example, several studies have employed reference-free attack, especially LOSS attack [73], for privacy auditing [31] or more sophisticated attack [6]. Mireshghallah et al. introduce the seminal reference-based attack, Likelihood Ratio Attacks (LiRA) [8, 70, 72], into Masked Language Models (MLMs), which measure the calibrated likelihood of a specific record by comparing the discrepancy on the likelihood between the target LLM and the reference LLM. Following this concept, Mireshghallah et al. further adapt LiRA for analyzing memorization in Causal Language Models (CLMs). However, these methods heavily rely on several over-optimistic assumptions, including assuming the overfitting of target LLMs [41] and having access to a reference dataset from the same distribution as the training dataset [46, 47]. Thus, it remains inconclusive whether prior MIAs can cause considerable privacy risk in practical scenarios.

As illustrated in Fig. 1, LiRA [47] and LOSS Attack [73] are employed to represent reference-based and reference-free MIAs to explore their performance in practical scenarios. Firstly, as shown in Fig. 1(a), we evaluate LiRA and LOSS Attack with three reference datasets from different sources, i.e., the dataset with the identical distribution with the member records (identical-distribution), the dataset of the same domain with the member records (domain-specific), and the dataset irrelevant to the member records (irrelevant). The performance of LOSS attack is consistently low and independent of the source of the reference dataset. For LiRA, the attack performance will catastrophically plummet as the similarity between the reference dataset and the target dataset declines. Thus, the reference-based MIA can not pose critical privacy leakage on LLMs since similar datasets are usually not available to adversaries in real applications. Secondly, as shown in Fig. 1(b), two target LLMs are fine-tuned over the same pre-trained model but stop before and after overfitting, and the reference LLMs are fine-tuned on a different dataset from the same domain. We can observe that existing MIAs cannot effectively cause privacy leaks when the LLM is not overfitting. This phenomenon is addressed by the fact that the membership signal proposed by existing MIAs is highly dependent on overfitting in

target LLMs. They assume that member records tend to have overall higher probabilities of being sampled than non-member ones, an assumption that is only satisfied in overfitting models [67].

In this work, to address the aforementioned two limitations of existing works, we propose a Membership Inference Attack based on Self-calibrated Probabilistic Variation (SPV-MIA) composed of two according modules. First, although existing reference-based MIAs are challenging to reveal actual privacy risks, they demonstrate the significant potential of achieving higher privacy risks with the reference model. Therefore, we design a self-prompt approach to extract the reference dataset by prompting the target LLMs themselves and collecting the generated texts. This approach allows us to acquire the significant performance improvement brought by the reference model while ensuring the adversary model is feasible on the practical LLMs. Second, prior studies have shown that memorization is intrinsic for machine learning models to achieve optimality [16] and can persist in LLMs without leading to overfitting [64]. Consequently, rather than relying on probabilities as membership signals, we propose designing a more resilient signal grounded in an insightful theory, which posits that LLM memorization manifests as an augmented concentration in the probability distribution surrounding the member records [67]. Specifically, we proposed a probabilistic variation metric that can detect local maxima points via the notion of second partial derivative test [62] approximately instantiated by a paraphrasing model. Moreover, based on the new theoretical foundation, we elucidate the efficacy of the neighbor attack [41] through the lens of LLM memorization. This analysis underscores the pivotal role of characterizing memorization in MIA for future studies. It is worth noting that our paraphrasing model does not rely on another MLM like the neighbour attack. Overall, our contributions are summarized as follows:

- We propose a self-prompt approach that collects reference datasets by prompting the target LLM to generate, which will have the closely resemble distribution as the training dataset. In this manner, the reference model fine-tuned on the reference dataset can significantly improve the attack performance without any unrealistic assumptions.
- We further design a probabilistic variation metric based on the theoretical foundation of LLM memorization [67], and derive a more convincing principle and explanation of the neighbour attack [41]. Furthermore, our investigation highlights the importance of characterizing LLM memorization for subsequent studies in designing more sophisticated MIA methods.
- We conducted extensive experiments to validate the effectiveness of SPV-MIA. The results suggest that SPV-MIA unveils significantly higher privacy risk across multiple fine-tuned LLMs and datasets compared with existing MIAs (about 23.6% improvement in AUC across four representative LLMs and three datasets).

## 2 Related Works

**Membership Inference Attack:** Initially, prior MIAs mainly focused on classical machine learning models, such as classification models [7, 10, 39, 60]. With the rapid development of other machine learning tasks, such as recommendation and generation tasks, MIAs against these task-specific models became a research direction of great value, and have been well investigated [13, 18, 77]. Meanwhile, ChatGPT released by OpenAI has propelled the attention towards LLMs to the peak over the past year, which promotes the study of MIAs against LLMs. The seminal works in this area typically focuses on fine-tuned LLM or LM. Mireshghallah et al. proposed LiRA against MLMs via adopting pre-trained models as reference models. Following this study, Mireshghallah et al. further adapted LiRA for CLMs. Mattern et al. pointed out the unrealistic assumption of a reference model trained on similar data, then substituted it with a neighbourhood comparison method. Although MIAs against LMs and fine-tuned LLMs have been studied by several works, the attack performance of existing MIAs in regard to LLMs with large-scale parameters and pre-trained on tremendous corpora is still not clear. Thus, some contemporaneous works have also been released to detect the pre-training data of LLMs and expose considerable privacy risks [11, 14, 19, 42, 59, 76]. In this work, we still focus on fine-tuned LLMs since the fine-tuning datasets are typically more private and sensitive. We evaluate previous MIAs on LLMs in practical scenarios, and found that the revealed privacy breaches were far below expectations due to their strict requirements and over-optimistic assumptions. Then, we propose SPV-MIA, which discloses significant privacy risks on practical LLM applications.

**Large Language Models:** In the past year, LLMs have dramatically improved performances on multiple natural language processing (NLP) tasks and consistently attracted attention in both academic

and industrial circles [44]. The widespread usage of LLMs has led to much other contemporaneous work on quantifying the privacy risks of LLMs [41, 48, 53]. In this work, we audit privacy leakages of LLMs by distinguishing whether or not a specific data record is used for fine-tuning the target LLM The existing LLMs primarily fall into three categories: causal language modeling (CLM) (e.g. GPT), masked language modeling (MLM) (e.g. BERT), and Sequence-to-Sequence (Seq2Seq) approach (e.g. BART). Among these LLMs, CLMs such as GPT [54, 69] and LLaMA [65] have achieved the dominant position with the exponential improvement of model scaling [79]. Therefore, we select CLM as the representative LLM for evaluation in this work.

## 3 Preliminaries

### 3.1 Causal Language Models

For a given text record $\boldsymbol{x}$, it can be split into a sequence of tokens $[t_0, t_1, \cdots, t_{|\boldsymbol{x}|}]$ with variable length $|\boldsymbol{x}|$. CLM is an autoregressive language model, which aims to predict the conditional probability $p_\theta\left(t_i \mid \boldsymbol{x}_{<i}\right)$ given the previous tokens $\boldsymbol{x}_{<i} = [t_0, t_1, \cdots, t_{i-1}]$. During the training process, CLM calculates the probability of each token in a text with the previous tokens, then factorizes the joint probability of the text into the product of conditional token prediction probabilities. Therefore, the model can be optimized by minimizing the negative log probability:

$$\mathcal{L}_{\text{CLM}} = -\frac{1}{M} \sum_{j=1}^{M} \sum_{i=1}^{\left|\boldsymbol{x}^{(j)}\right|} \log p_\theta\left(t_i \mid \boldsymbol{x}_{<i}^{(j)}\right), \tag{1}$$

where $M$ denotes the number of training records. In the process of generation, CLMs can generate coherent words by predicting one token at a time and producing a complete text using an autoregressive manner. Moreover, the pretraining-finetuning paradigm is proposed to mitigate the uncountable demands of training an LLM for a specific task [44]. Besides, multifarious parameters-efficient fine-tuning methods (e.g., LoRA [24], P-Tuning [38]) are introduced to further decrease consumption by only fine-tuning limited model parameters [12]. In this work, we concentrate on the fine-tuning phase, since the fine-tuning datasets are usually more private and vulnerable to the adversary [74].

### 3.2 Threat Model

In this work, we consider an adversary who aims to infer whether a specific text record was included in the fine-tuning dataset of the target LLM. There are two mainstream scenarios investigated by previous research: white-box and black-box MIAs. White-box MIA assumes full access to the raw copy of the target model, which means the adversary can touch and modify each part of the target model [50]. For a fully black-box scenario, the adversary should only approved to acquire output texts generated by the target LLM while given specific prompts, which maybe too strict for the adversary to conduct a valid MIA [14, 41, 58, 76]. Thus, we consider a practical setting beyond fully black-box that further requires two regular API access for evaluating existing works and our proposed method:

- **Query API**: The access to the query API that only provides generated texts and logits (or loss).
- **Fine-tuning API**: The access to the fine-tuning API of the pre-trained version of the target model.

Note that the query API access is widely adopted by existing MIA works [14, 41, 58, 76], both the query API and the fine-tuning API are usually provided by commercial LLM providers, such as OpenAI [52] and Zhipu AI [2]. These two APIs are also easy for the adversary to access in open-source LLMs, such as LLaMA [65] and Flacon [3]. In our setting, $D$ is a dataset collected for a specific task, which can be separated into two disjoint subsets: $D_{mem}$ and $D_{non}$. The target LLM $\theta$ is fine-tuned on $D_{mem}$, and the adversary has no prior information about which data records are utilized for fine-tuning. Besides, all reference-based MIA, including SPV-MIA, can at most fine-tune the reference model using a disjoint dataset $D_{refer}$ from the same task. The adversary algorithm $\mathcal{A}$ is designed to infer whether a text record $\boldsymbol{x}^{(i)} \in D$ belong to the training dataset $D_{mem}$:

$$\mathcal{A}\left(\boldsymbol{x}^{(j)}, \theta\right) = \mathbb{1}\left[P\left(m^{(j)} = 1 | \boldsymbol{x}^{(j)}, \theta\right) \geq \tau\right], \tag{2}$$

where $m^{(j)} = 1$ indicates that the record $\boldsymbol{x}^{(j)} \in D_{mem}$, $\tau$ represents the threshold, and $\mathbb{1}$ denotes the indicator function.

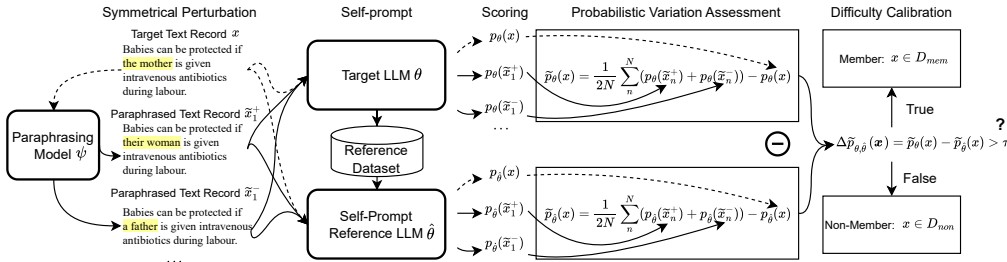

Figure 2: The overall workflow of SPV-MIA, where includes the probabilistic calibration via self-prompt reference model and the probabilistic variation assessment via paraphrasing model.

## 4 Membership Inference Attack via Self-calibrated Probabilistic Variation

In this section, we first introduce the general paradigm of Membership Inference Attack via Self-calibrated Probabilistic Variation (SPV-MIA) as illustrated in Fig 2. Then we discuss the detailed algorithm instantiations of this general paradigm by introducing practical difficulty calibration (PDC, refer to Section 4.2) and probabilistic variation assessment (PVA, refer to Section 4.3).

### 4.1 General Paradigm

As formulated in Eq. 1, the objective of an LLM is to maximize the joint probability of the text in the training set. Thus, prior **reference-free MIAs** employ the joint probability of the target text being sampled as the membership signal [6, 31, 58]:

$$\mathcal{A}\left(\boldsymbol{x},\theta\right) = \mathbb{1}\left[p_\theta\left(\boldsymbol{x}\right) \geq \tau\right], \tag{3}$$

where $p_\theta\left(\boldsymbol{x}\right)$ denotes the probability measured on the target model $\theta$. Since some records are inherently over-represented, even non-member records can achieve high probability in the data distribution [70], which leads to a high False-Positive Rate (FPR). Thus, **reference-based MIAs** adopt difficulty calibration [14, 57, 70], which further calibrates the probability by comparing it with the value measured on reference models [46, 47]:

$$\mathcal{A}_{exist}\left(\boldsymbol{x},\theta\right) = \mathbb{1}\left[\Delta p_\theta\left(\boldsymbol{x}\right) \geq \tau\right] = \mathbb{1}\left[p_\theta\left(\boldsymbol{x}\right) - p_\phi\left(\boldsymbol{x}\right) \geq \tau\right], \tag{4}$$

where $\Delta p_\theta\left(\boldsymbol{x}\right)$ is the calibrated probability, and $p_\phi\left(\boldsymbol{x}\right)$ is estimated on the reference model $\phi$.

However, both reference-free and reference-based MIAs often encounter high FPR in practical scenarios [14, 41, 59, 76]. For reference-based MIAs, although them has the potential to offset the over-represented statuses of data records if the reference model can be trained on a dataset closely resembling the training dataset $D_{mem}$. Nevertheless, it is almost unrealistic for an adversary to obtain such a dataset, and adopting a compromising dataset will lead to the collapse of attack performance. We circumvent this by introducing a self-prompt reference model $\widehat{\theta}$, which is trained on the generated text of the target model $\theta$. Besides, the probability signal adopted by existing MIAs is not reliable, since the confidence of the probability signal is notably declined when the target model is not overfitting only memorization [67]. Thus, we elaborately design a more stable membership signal, probabilistic variation $\widetilde{p}_\theta\left(\boldsymbol{x}\right)$, audited by a paraphrasing model and only rely on LLM memorization. Formally, as depicted in Fig. 2, **our proposed MIAs** can be formulated as:

$$\mathcal{A}_{our}\left(\boldsymbol{x},\theta,\widehat{\theta}\right) = \mathbb{1}\left[\Delta\widetilde{p}_{\theta,\widehat{\theta}}\left(\boldsymbol{x}\right) \geq \tau\right] = \mathbb{1}\left[\widetilde{p}_\theta\left(\boldsymbol{x}\right) - \widetilde{p}_{\widehat{\theta}}\left(\boldsymbol{x}\right) \geq \tau\right], \tag{5}$$

where $\widetilde{p}_\theta\left(\boldsymbol{x}\right)$ and $\widetilde{p}_{\widehat{\theta}}\left(\boldsymbol{x}\right)$ are probabilistic variations of the text record $\boldsymbol{x}$ measured on the target model $\theta$ and the self-prompt reference model $\widehat{\theta}$ respectively.

### 4.2 Practical Difficulty Calibration (PDC) via Self-prompt Reference Model

Watson et al. [70] has suggested that inferring the membership of a record by thresholding on a predefined metric (e.g. confidence [56], loss [73], and gradient norm [50]) will cause a high FPR. Since several non-member records may have high probabilities of being classified as member records simply because they are inherently over-represented in the data manifold. In other words, the metric estimated on the target model is inherently biased and has a high variance, which leads to a

significant overlap in the metric distributions between members and non-members, making them more indistinguishable. To mitigate this phenomenon, Watson et al. propose difficulty calibration as a general approach for extracting a much more distinguishable membership signal, which can be adapted to most metric-based MIAs by constructing their calibrated variants [46, 46, 70]. Concretely, difficulty calibration assumes an ideal reference dataset $D_{refer}$ drawn from the identical distribution as the training set $D_{mem}$ of the target model $\theta$, and trains an ideal reference model $\phi$ with a training algorithm $\mathcal{T}$. Then, it fabricates a calibrated metric by measuring the discrepancy between metrics on the target model and reference model, and this can offset biases on membership signals caused by some over-represented records. The calibrated metric is defined as:

$$\Delta m(\boldsymbol{x}) = m_\theta(\boldsymbol{x}) - \mathbb{E}_{\phi \leftarrow \mathcal{T}(\mathcal{D}_{refer})}[m_\phi(\boldsymbol{x})], \tag{6}$$

where $\Delta m(\boldsymbol{x})$ is the calibrated metric, $m_\theta(\boldsymbol{x})$ and $m_\phi(\boldsymbol{x})$ are metrics measured on target and reference models, respectively. The existing study has verified that reference-based MIA highly depends on the similarity of training and reference dataset [41]. A low-quality dataset will lead to an exponential decrease in attack performance. However, the dataset used for fine-tuning an LLM is typically highly private, making extracting a high-quality reference dataset from the same distribution a non-trivial challenge.

We notice that LLMs possess revolutionary fitting and generalization capabilities, enabling them to generate a wealth of creative texts. Therefore, LLMs themselves have the potential to depict the distribution of the training data. Thus, we consider a self-prompt approach that collects the reference dataset from the target LLM itself by prompting it with few words. Concretely, we first collect a set of text chunks with an equal length of $l$ from a public dataset from the same domain, where the domain can be easily inferred from the task of the target LLM (e.g., An LLM that serves to summary task has high probability using a summary fine-tuning dataset). Then, we utilize each text chunk of length $l$ as the prompt text and request the target LLM to generate text. All the generated text can form a dataset of size $N$, which is used to fine-tune the proposed self-prompt reference model $\widehat{\theta}$ over the pre-trained model. Accordingly, we can define the practical difficulty calibration as:

$$\Delta m(\boldsymbol{x}) = m_\theta(\boldsymbol{x}) - \mathbb{E}_{\widehat{\theta} \leftarrow \mathcal{T}(\mathcal{D}_{self})}[m_{\widehat{\theta}}(\boldsymbol{x})] \approx m_\theta(\boldsymbol{x}) - m_{\widehat{\theta}}(\boldsymbol{x}), \tag{7}$$

where $\mathcal{D}_{self} \sim p_\theta(\boldsymbol{x})$, $m_\theta(\boldsymbol{x})$ and $m_{\widehat{\theta}}(\boldsymbol{x})$ are membership metrics measured over the target model and the self-prompt reference model. Only one reference model is used for computational efficiency, which can achieve sufficiently high attack performance. It is worth noting that in some challenging scenarios where acquiring domain-specific datasets is difficult, our self-prompt method can still effectively capture the underlying data distribution, even when using completely unrelated prompt texts. The relevant experiments will be conducted and discussed in detail in Section 5.4.

### 4.3 Probabilistic Variation Assessment (PVA) via Symmetrical Paraphrasing

Before diving into technical details, we first provide a brief overview of the motivation behind our proposed probabilistic variation assessment by demonstrate that memorization is a more reliable membership signal. Although memorization is associated with overfitting, overfitting by itself cannot completely explain some properties of memorization[47, 64, 75]. The key differences between memorization and overfitting can be summarized as the following three points:

- **Occurrence Time:** Existing research defines the first epoch when the LLM's perplexity (PPL) on the validation set starts to rise as the occurrence of overfitting [64]. In contrast, memorization begins early [47, 64] and persists throughout almost the entire training phase [47, 75].

- **Harm Level:** Overfitting is almost universally acknowledged as a detrimental phenomenon in machine learning. However, memorization is not exclusively harmful, and can be crucial for certain types of generalization (e.g., on QA tasks) [4, 63].

- **Avoidance Difficulty:** Since memorization occurs much earlier, even if we use early stopping to prevent overfitting, we will still achieve significant memorization [47]. Memorization has been verified as an inevitable phenomenon for achieving optimal generalization on machine learning models [17]. Moreover, since memorization is crucial for certain LLM tasks [4, 63], and separately mitigates specific unintended memorization (e.g., verbatim memorization [27]) is a non-trivial task.

Therefore, the aforementioned discussions highlights memorization will naturally be a more reliable signal for detecting member text. Memorization in generative models will cause member records to

have a higher probability of being generated than neighbour records in the data distribution [67]. This principle can be shared with LLMs, as they can be considered generation models for texts. Thus, we suggest designing a more promising membership signal that can measure a value for each text record to identify whether this text is located on the local maximum in the sample distribution characterized by $\theta$. The second partial derivative test is an approach in multivariable calculus commonly employed to ascertain whether a critical point of a function is a local minimum, maximum, or saddle point [62]. For our objective of identifying maximum points, we need to confirm if the Hessian matrix is negative definite, meaning that all the directional second derivatives are negative. However, considering that member records may not strictly fall on maximum points, we suggest relaxing the decision rule and using specific statistical metrics of the distribution of the second-order directional derivative over the direction $z$ to characterize the probability variation. Thus, we define the probabilistic variation mentioned in Eq. 5 as the expectation of the directional derivative:

$$\widetilde{p}_\theta\left(\boldsymbol{x}\right) := \mathbb{E}_{\boldsymbol{z}}\left(\boldsymbol{z}^\top H_p\left(\boldsymbol{x}\right)\boldsymbol{z}\right), \tag{8}$$

where $H_p(\cdot)$ is the hessian matrix of the probability function $p_\theta(\cdot)$, then $\boldsymbol{z}^\top H_p\left(\boldsymbol{x}\right)\boldsymbol{z}$ indicates the second-order directional derivative of $p_\theta(\cdot)$ with respect to the text record $x$ in the direction $\boldsymbol{z}$. However, calculating the second-order derivative is computationally expensive and may not be feasible in LLMs. Thus, we propose a practical approximation method to evaluate the probabilistic variation. Specifically, we further approximate the derivative with the symmetrical form [26]:

$$\boldsymbol{z}^\top H_p(\boldsymbol{x})\boldsymbol{z} \approx \frac{p_\theta(\boldsymbol{x}+h\boldsymbol{z}) + p_\theta(\boldsymbol{x}-h\boldsymbol{z}) - 2p_\theta(\boldsymbol{x})}{h^2}, \tag{9}$$

where requires $h \to 0$, and $z$ can be considered as a sampled perturbation direction. Thus, $\boldsymbol{x} \pm h\boldsymbol{z}$ can be considered as a pair of symmetrical adjacent text records of $\boldsymbol{x}$ in the data distribution. Then we can reformulate Eq. 8 as follows by omitting coefficient $h$:

$$\widetilde{p}_\theta\left(\boldsymbol{x}\right) \approx \frac{1}{2N}\sum_n^N \left(p_\theta\left(\widetilde{\boldsymbol{x}}_n^+\right) + p_\theta\left(\widetilde{\boldsymbol{x}}_n^-\right)\right) - p_\theta\left(\boldsymbol{x}\right). \tag{10}$$

where $\widetilde{\boldsymbol{x}}_n^\pm = \boldsymbol{x} \pm \boldsymbol{z}_n$ is a symmetrical text pair sampled by a paraphrasing model, which slightly paraphrases the original text $\boldsymbol{x}$ in the high-dimension space. Note that the paraphrasing in the sentence-level should be modest as Eq. 9 requires $h \to 0$, but large enough to ensure enough precision to distinguish the probabilistic variation in Eq. 8. Based on the aforementioned discussions, we designed two different paraphrasing models in the embedding domain and the semantic domain, respectively, to generate symmetrical paraphrased text embeddings or texts. For the embedding domain, we first embed the target text, then randomly sample noise following Gaussian distribution, and obtain a pair of symmetrical paraphrased texts by adding/subtracting noise. For the semantic domain, we randomly mask out 20% tokens in each target text, then employ T5-base to predict the masked tokens. Then, we compute the difference in the embeddings between the original tokens and predicted tokens to search for tokens that are symmetrical to predicted tokens with respect to the original tokens. We provide the detailed pseudo codes of both two paraphrasing models in Appendix A.3. In subsequent experiments, we default to paraphrasing in the semantic domain. Furthermore, we reformulate the neighbour attack and provide another explanation of its success based on the probabilistic variation metric with a more rigorous principle (refer to Appendix A.4). Additionally, supplementary experiments demonstrate that our proposed paraphrasing model in the embedding domain achieves considerable performance gains without relying on another MLM.

## 5 Experiments

### 5.1 Experimental Setup

Our experiments are conducted on four open-source LLMs: **GPT-2** [54], **GPT-J** [69], **Falcon-7B** [3] and **LLaMA-7B** [65], which are both fine-tuned over three dataset across multiple domains and LLM use cases: **Wikitext-103** [43], **AG News** [78] and **XSum** [49]. Each target LLM is fine-tuned with the batch size of 16, and trained for 10 epochs. Each self-prompt reference model is trained for 4 epochs. We adopt LoRA [24] as the default Parameter-Efficient Fine-Tuning (PEFT) technique. The learning rate is set to 0.0001. We adopt the AdamW optimizer [40] and early stopping [71] to avoid overfitting and achieve generalization in LLMs, the PPL of each LLM-dataset pair is provided in Appendix A.5.4. We compare SPV-MIA with seven state-of-the-art MIAs

| Method | Wiki | | | | | AG News | | | | | Xsum | | | | |
|---|---|---|---|---|---|---|---|---|---|---|---|---|---|---|---|
| | GPT-2 | GPT-J | Falcon | LLaMA | Avg. | GPT-2 | GPT-J | Falcon | LLaMA | Avg. | GPT-2 | GPT-J | Falcon | LLaMA | Avg. |
| Loss Attack | 0.614 | 0.577 | 0.593 | 0.605 | 0.597 | 0.591 | 0.529 | 0.554 | 0.580 | 0.564 | 0.628 | 0.564 | 0.577 | 0.594 | 0.591 |
| Neighbour Attack | 0.647 | 0.612 | 0.621 | 0.627 | 0.627 | 0.622 | 0.587 | 0.594 | 0.610 | 0.603 | 0.612 | 0.547 | 0.571 | 0.582 | 0.578 |
| DetectGPT | 0.623 | 0.587 | 0.603 | 0.619 | 0.608 | 0.611 | 0.579 | 0.582 | 0.603 | 0.594 | 0.603 | 0.541 | 0.563 | 0.577 | 0.571 |
| Min-K% | 0.658 | 0.623 | 0.629 | 0.643 | 0.638 | 0.629 | 0.604 | 0.607 | 0.619 | 0.615 | 0.621 | 0.562 | 0.588 | 0.594 | 0.591 |
| Min-K%++ | 0.623 | 0.613 | 0.645 | 0.648 | 0.635 | 0.635 | 0.609 | 0.623 | 0.631 | 0.625 | 0.627 | 0.556 | 0.589 | 0.604 | 0.594 |
| LiRA-Base | 0.710 | 0.681 | 0.694 | 0.709 | 0.699 | 0.658 | 0.634 | 0.641 | 0.657 | 0.648 | 0.776 | 0.718 | 0.734 | 0.759 | 0.747 |
| LiRA-Candidate | 0.769 | 0.726 | 0.735 | 0.748 | 0.744 | 0.717 | 0.690 | 0.708 | 0.714 | 0.707 | 0.823 | 0.772 | 0.785 | 0.809 | 0.797 |
| SPV-MIA | **0.975** | **0.929** | **0.932** | **0.951** | **0.938** | **0.949** | **0.885** | **0.898** | **0.903** | **0.909** | **0.944** | **0.897** | **0.918** | **0.937** | **0.924** |

Table 1: **AUC Score** for detecting member texts from four LLMs across three datasets for SPV-MIA and five previously proposed methods. **Bold** and Underline respectively represent the best and the second-best results within each column (model-dataset pair).

| Method | Wiki | | | | | AG News | | | | | Xsum | | | | |
|---|---|---|---|---|---|---|---|---|---|---|---|---|---|---|---|
| | GPT-2 | GPT-J | Falcon | LLaMA | Avg. | GPT-2 | GPT-J | Falcon | LLaMA | Avg. | GPT-2 | GPT-J | Falcon | LLaMA | Avg. |
| Loss Attack | 1.3% | 1.2% | 1.1% | 1.4% | 1.2% | 1.3% | 1.0% | 1.3% | 1.2% | 1.2% | 1.5% | 1.0% | 1.0% | 1.1% | 1.2% |
| Neighbour Attack | 4.1% | 3.6% | 2.8% | 3.4% | 3.5% | 3.6% | 2.7% | 2.8% | 3.1% | 3.1% | 3.2% | 2.4% | 2.5% | 2.7% | 2.7% |
| DetectGPT | 3.7% | 3.1% | 2.6% | 3.2% | 3.2% | 3.3% | 2.4% | 2.6% | 2.7% | 2.8% | 3.0% | 2.1% | 2.4% | 2.6% | 2.5% |
| Min-K% | 4.4% | 4.3% | 3.4% | 3.7% | 4.0% | 3.7% | 3.4% | 3.8% | 3.6% | 3.6% | 3.4% | 2.5% | 2.7% | 3.1% | 2.9% |
| Min-K%++ | 3.7% | 4.2% | 3.8% | 3.9% | 3.9% | 4.0% | 3.3% | 3.9% | 4.1% | 3.8% | 3.1% | 2.8% | 3.2% | 3.4% | 3.1% |
| LiRA-Base | 12.5% | 11.3% | 10.7% | 11.2% | 11.4% | 9.2% | 8.0% | 8.3% | 8.7% | 8.6% | 13.5% | 9.3% | 10.7% | 12.2% | 11.4% |
| LiRA-Candidate | 16.3% | 14.3% | 14.8% | 15.0% | 15.1% | 12.2% | 9.4% | 10.6% | 11.5% | 10.9% | 19.4% | 10.9% | 14.5% | 18.5% | 15.8% |
| SPV-MIA | **67.3%** | **55.4%** | **57.6%** | **64.2%** | **61.1%** | **42.9%** | **34.8%** | **37.6%** | **39.5%** | **38.7%** | **42.1%** | **38.6%** | **40.7%** | **42.0%** | **40.9%** |

Table 2: **TPR@1%FPR** for detecting member texts from four LLMs across three datasets for SPV-MIA and five previously proposed methods.

designed for LMs, including five reference-free MIAs: **Loss Attack** [73], **Neighbour Attack** [41], **DetectGPT** [48], **Min-K%** [59], **Min-K%++** [76] and two reference-based MIAs: **LiRA-Base** [47], **LiRA-Candidate** [47]. We defer the detailed setup information to Appendix A.6.

## 5.2 Overall Performance

As presented in Table 1, we initially summarize the AUC scores [5] for all baselines and SPV-MIA against four LLMs across three datasets. Then, as illustrated in Tabel 2, we follow the suggestion of Carlini et al. [7] to evaluate the MIA performance by computing True-Positive Rate (TPR) at low False-Positive Rate (FPR). Furthermore, we present linear scale and logarithmic scale receiver operating characteristic (ROC) curves for SPV-MIA and the top three representative baselines on LLaMAs in Appendix A.5.3 for a more comprehensive presentation. The results demonstrate that SPV-MIA achieves the best overall attack performance with the highest average AUC of $0.924$ over all scenarios. In comparison to the AUC score, SPV-MIA demonstrates a more substantial performance margin TPR at low FPR, achieving an average TPR@1% FPR of 46.9%. Furthermore, compared to the most competitive baseline, LiRA-Canididate, SPV-MIA has improved the AUC of the attack by $30\%$, even LiRA-Canididate assumes full access to the auxiliary dataset while SPV-MIA only needs some short text chunks from this dataset. This phenomenon indicates that our proposed self-prompt approach enables the reference model to gain a deeper understanding of the data distribution, thereby serving as a more reliable calibrator. We also conduct ablation studies to evaluate the contribution of both PDC and PVA in SPV-MIA, and the results are presented in Appendix A.5.1. Most baseline, especially reference-free attack methods, yield a low AUC, which is only slightly better than random guesses. Furthermore, their performances on larger-scale LLMs are worse. This phenomenon verifies the claim that existing MIAs designed for LMs can not handle LLMs with large-scale parameters. It is also worth noting that the privacy risks caused by MIAs are proportional to the overall parameter scale and language capabilities of LLMs. We interpret this phenomenon as follows: LLMs with stronger overall NLP performance have better learning ability, which means they are more likely to memorize records from the training set. Besides, MIAs fundamentally leverage the memorization abilities of machine learning models, making superior models more vulnerable to attacks.

## 5.3 How MIAs Rely on Reference Dataset Quality

In this work, a key contribution is introducing a self-prompt approach for constructing a high-quality dataset to fine-tune the reference model, which guides the reference model to become a better calibrator. Therefore, we conduct experiments to investigate how prior reference-based MIAs rely on the quality of the reference dataset, and evaluate whether our proposed method can build a high-

quality reference dataset. In real-world scenarios, based on different prior information, adversaries can obtain datasets from different sources to fine-tune the reference model with uneven quality.

We categorize the reference dataset into three types based on their relationship with the fine-tuning dataset of the target model and sort them in ascending order of difficulty in acquisition: 1) **Irrelevant** dataset, 2) **Domain-specific** dataset, and 3) **Identical distribution** dataset. Besides, the dataset extracted by the self-prompt approach is denoted as 4) **Self-prompt** dataset. The detailed information of these datasets is summarized in Appendix A.6. Then, we conduct MIAs with the aforementioned four data sources and summarize the results in Fig. 3. The experimental results indicate that the performance of MIA shows a noticeable decrease along the Identical, Domain, and Irrelevant datasets. This illustrates the high dependency of previous reference-based methods on the quality of the reference

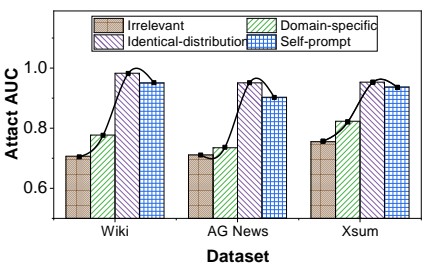

Figure 3: The performances of reference-based MIA on LLaMA while utilizing different reference datasets.

dataset. However, AUC scores on self-prompt reference datasets are only marginally below Identical datasets. It verifies that our proposed self-prompt method can effectively leverage the creative generation capability of LLMs, approximate sampling high-quality text records indirectly from the distribution of the target training set.

## 5.4 The Robustness of SPV-MIA in Practical Scenarios

We have verified that SPV-MIA can provide a high-quality reference model. However, the source and scale of self-prompt texts may face various limitations in practical scenarios. Therefore, we conducted experiments to verify the robustness of SPV-MIA performance in diverse practical scenarios. **Source of Self-prompt Texts.** The sources of self-prompt texts available to attackers are usually limited by the actual deployment environment, and sometimes even domain-specific texts may not be accessible. Compared with using domain-specific text chunks for prompting, we also evaluate the self-prompt approach with irrelevant and identical-distribution text chunks. As shown in Fig. 4, the self-prompt method demonstrates an incredibly lower dependence on the source of the prompt texts. We

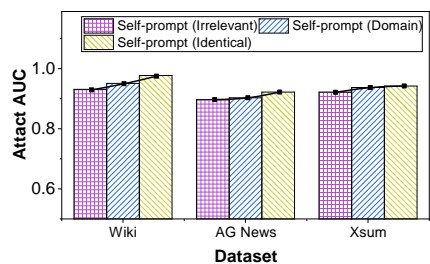

Figure 4: The performances of SPV-MIA on LLaMA while utilizing different prompt text sources.

found that even when using completely unrelated prompt texts, the performance of the attack only experiences a slight decrease (3.6% at most). This phenomenon indicates that the self-prompt method we proposed has a high degree of versatility across adversaries with different prior information.

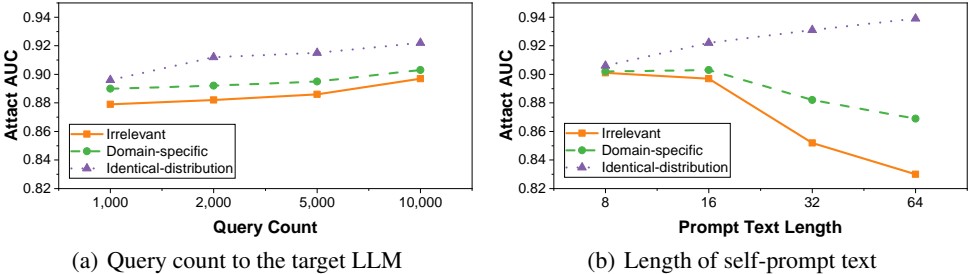

(a) Query count to the target LLM

(b) Length of self-prompt text

Figure 5: The performances of SPV-MIA on LLaMA while utilizing different query count to the target model and different prompt text lengths.

**Scale of Self-prompt Texts.** In real-world scenarios, the scale of self-prompt texts is usually limited by the access frequency cap of the LLM API and the number of available self-prompt texts. Thus, we set up two sets of experiments to verify the sensitivity of SPV-MIA to the aforementioned limitations. As shown in Fig. 5(a), our self-prompt reference model is minimally affected by the access frequency limitations of the target LLM. Even with only 1,000 queries, it achieves performance comparable to

10,000 queries. As shown in Fig. 5(b), even when the self-prompt texts are severely limited (with only 8 prompt tokens), the attack performance remains at a startlingly high level of 0.9. Besides, texts from different sources show varying attack performance trends based on text length. From the identical dataset, attack performance increases with text length. From the domain-specific dataset, it initially increases then decreases. From an irrelevant dataset, it decreases with longer texts. Therefore, we recommend setting smaller text lengths to allow LLMs to generate samples that are close to data distributions of training sets, unless adversaries can directly sample texts from the same data distribution as the training set. Overall, our proposed method can maintain stable attack performance in practical scenarios where the scale of self-prompt texts is limited.

## 5.5 Defending against SPV-MIAs

As privacy risks emerge from various attacks, including data extraction attack [6], model extraction attack [22], and membership inference attack [41, 60, 72], the research community actively promotes defending methods against these attacks [29, 45]. DP-SGD [1] is one of the most widely adopted defense methods based on differential privacy [15] to provide mathematical privacy guarantees. Through DP-SGD, the amount of information the parameters have about a single data record is bound. Thus, the privacy leakage will not exceed the upper bound, regardless of

| Privacy Budget $\epsilon$ | 15 | 30 | 60 | $+\inf$ |
|---|---|---|---|---|
| Wiki | 0.785 | 0.832 | 0.875 | 0.951 |
| AG News | 0.766 | 0.814 | 0.852 | 0.903 |
| Xsum | 0.771 | 0.827 | 0.867 | 0.937 |
| Avg. | 0.774 | 0.824 | 0.865 | 0.930 |

Table 3: The AUC performance of SPV-MIA against LLaMA fine-tuned with DP-SGD w.r.t different privacy budget $\epsilon$.

how many outputs we obtain from the target model. We follow the same manner as the existing study [36] and train LLaMA with DP-Adam on the three datasets. The results are summarized in Table 3, where we choose a set of appropriate $\epsilon$ as existing works suggest that higher DP guarantees lead to a noticeable performance degradation [21, 41]. The performances of LLMs are supplemented in Appendix A.5.4, and the results of other baselines can be found in Appendix A.5.5. The results indicate that DP-SGD can mitigate privacy risks to a certain extent. However, under moderate privacy budgets, SPV-MIA still presents a notable risk of privacy leakage and outperforms the baselines.

### 5.5.1 Impact of Fine-tuning Methods

We further evaluated the generalizability of SPV-MIA under different PEFT techniques. As shown in Table 4, SPV-MIA can maintain a high-level AUC across all PEFT techniques. Besides, the performance of MIA is positively correlated with the number of trainable parameters during the fine-tuning process. We hypothesize that this is because as the number of trainable parameters increases, LLMs retain more complete memory of the member records, making them more vulnerable to attacks.

| PEFT | LoRA | Prefix Tuning | P-Tuning | (IA)$^3$ |
|---|---|---|---|---|
| # Parameters (M) | 33.55 | 5.24 | 1.15 | 0.61 |
| Wiki | 0.951 | 0.943 | 0.922 | 0.914 |
| Ag News | 0.903 | 0.897 | 0.879 | 0.873 |
| Xsum | 0.937 | 0.931 | 0.924 | 0.911 |

Table 4: The AUC Performance of SPV-MIA across LLaMAs fine-tuned with different PEFT techniques over three datasets. We choose LoRA [24], Prefix Tuning [35], P-Tuning [38] and (IA)$^3$ [37] as four representative PEFT techniques.

## 6  Conclusion

In this paper, we reveal the under-performances of existing MIA methods against LLMs for practical applications and interpret this phenomenon from two perspectives. First, reference-based attacks seem to pose impressive privacy leakages by comparing the sampling probabilities of the target record between target and reference LLMs, but the inaccessibility of the appropriate reference dataset will be a big obstacle to deploying it in practice. Second, existing MIAs heavily rely on overfitting, which is usually avoided before releasing LLM for public access. To ddress these limitations, we propose a Membership Inference Attack based on Self-calibrated Probabilistic Variation (SPV-MIA), where we propose a self-prompt approach to extract reference dataset from LLM itself in a practical manner, then introduce a more reliable membership signal based on memorization rather than overfitting. We conduct substantial experiments to validate the superiority of SPV-MIA over all baselines and verify its effectiveness in extreme conditions. One primary limitation of this study is that SPV-MIA is only designed for CLM, we leave the adaption on other LLMs as the future work.

## Acknowledgments

This research was supported in part by the National Natural Science Foundation of China under Grants U21B2036, U23B2030, 62272262 and the Postdoctoral Fellowship Program of CPSF under Grant Number GZC20240548. We are also grateful for the support from the 6G Research Center, HUST, and the School of Cyber Science and Engineering, HUST, in the form of an International Travel Grant, which enabled Wenjie Fu to attend the conference in person.

We would like to express our sincere gratitude to the anonymous reviewers for their insightful comments and suggestions. We also extend our thanks to Jamie Hayes at Google DeepMind and Qilong Zhang at ByteDance for their valuable feedback on the preprint manuscript.

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

# A Appendix

## A.1 Ethic and Broader Impact Statements

For ethic consideration, this study proposes a membership inference attack algorithm, SPV-MIA, which can be maliciously utilized to infer whether a specific textual entry is fed to the target LLM during the training process. The extensive experiments reveal appreciable privacy leakage of LLMs through SPV-MIA, where the member records are identified with high confidence. We acknowledge that SPV-MIA can bring severe privacy risks to existing LLMs. Therefore, to prevent potential misuse of this research, all experimental findings are based on widely used public datasets. This ensures that every individual textual record we analyze has already been made public, and eliminates any further privacy violations.

For broader impact, we have made our code accessible to the public to allow additional research in the pursuit of identifying appropriate defense solutions. Thus, we posit that our article can inspire forthcoming research to not only focus on the linguistic ability of LLMs, but also take into account the dimensions of public data privacy and security. Besides, our research is more closely aligned with real-world LLM application scenarios, thereby revealing privacy risks in more realistic settings. Our proposed method is scalable, with many aspects left for further exploration and research, such as adapting it for other LLMs.

## A.2 Notations of This Work

Table 5: Notations and descriptions.

| Notation | Description |
|---|---|
| $\boldsymbol{x}$ | A specific data record. |
| $\widetilde{\boldsymbol{x}}_n^{\pm}$ | A pair of symmetrical paraphrasing text record of the target text record $\boldsymbol{x}$. |
| $D_{mem}$ | The training dataset utilized for LLM fine-tuning. |
| $D_{non}$ | A disjoint dataset from the training dataset. |
| $D_{refer}$ | The reference dataset that collected for fine-tuning reference LLM. |
| $m^{(j)}$ | The membership of the data record $\boldsymbol{x}^{(j)}$, 1 represents member, whereas 0 represents non-member. |
| $\theta$ | The parameters of the target large language model (LLM). |
| $\phi$ | The parameters of the reference LLM. |
| $\widehat{\theta}$ | The parameters of the self-prompt reference LLM. |
| $\mathcal{A}(\boldsymbol{x}, \theta)$ | The adversary algorithm for MIA. |
| $p_\theta(\boldsymbol{x})$ | The probability of text record $\boldsymbol{x}$ being sampled by the LLM $\theta$. |
| $p_\theta(\widetilde{\boldsymbol{x}}_n)$ | The probability of paraphrasing text $\widetilde{\boldsymbol{x}}_n$ being sampled by the LLM $\theta$. |
| $\Delta p_\theta(\boldsymbol{x})$ | The calibrated probability of text record $\boldsymbol{x}$. |
| $\widetilde{p}_\theta(\boldsymbol{x})$ | The probabilistic variation of $\boldsymbol{x}$ measured on the target LLM $\theta$. |
| $\widetilde{p}_{\widehat{\theta}}(\boldsymbol{x})$ | The probabilistic variation of $\boldsymbol{x}$ measured on the self-prompt reference LLM $\widehat{\theta}$. |
| $\Delta \widetilde{p}_{\theta,\widehat{\theta}}(\boldsymbol{x})$ | The calibrated probabilistic variation of $\boldsymbol{x}$ measured on both the target LLM $\theta$ and the self-prompt reference LLM $\widehat{\theta}$. |
| $N$ | The query times for estimating $\widetilde{p}_\theta(\boldsymbol{x})$. |

## A.3 Detailed Pseudo Codes of Symmetrical Perturbation

---

**Algorithm 1** Symmetrical paraphrase in the embedding domain

---

**Input:** Target text set $\{\boldsymbol{x}^{(i)}\}$, Gaussian noise scale $\sigma$, paraphrasing number $N$, embedding matrix of tokens $\mathbf{E}$.

**Ouput:** Symmetrical paraphrased text embedding $emb(\boldsymbol{x}^{(i)})^{\pm}$.

1: **for** $\boldsymbol{x}^{(i)} \in \{\boldsymbol{x}^{(i)}\}$ **do**
2:      $id(\boldsymbol{x}^{(i)}) \leftarrow tokenizer(\boldsymbol{x}^{(i)})$                ▷ Tokenize the text into token ids.
3:      $emb(\boldsymbol{x}^{(i)}) \leftarrow \mathbf{E}(id(\boldsymbol{x}^{(i)}))$            ▷ Convert token ids into embeddings.
4:      **for** $n \in \{1, \cdots, N\}$ **do**
5:          $\boldsymbol{z} \sim \mathcal{N}(0, \sigma^2 \boldsymbol{I})$           ▷ Sample noise from Gaussian distribution.
6:          $emb(\boldsymbol{x}^{(i)})^{+} \leftarrow emb(\boldsymbol{x}^{(i)}) + \boldsymbol{z}$
7:          $emb(\boldsymbol{x}^{(i)})^{-} \leftarrow emb(\boldsymbol{x}^{(i)}) - \boldsymbol{z}$
8:          **return** $emb(\boldsymbol{x}^{(i)})^{\pm}$
9:      **end for**
10: **end for**

---

**Algorithm 2** Symmetrical paraphrase in the semantic domain

---

**Input:** Target text set $\{\boldsymbol{x}^{(i)}\}$, paraphrasing percentage $\lambda$, paraphrasing number $N$, embedding matrix of tokens $\mathbf{E}$.

**Ouput:** Symmetrical paraphrased text $\boldsymbol{x}^{(i)\pm}$.

1: **for** $\boldsymbol{x}^{(i)} \in \{\boldsymbol{x}^{(i)}\}$ **do**
2:      $id(\boldsymbol{x}^{(i)}) \leftarrow tokenizer.encode(\boldsymbol{x}^{(i)})$         ▷ Tokenize the text into token ids.
3:      **for** $n \in \{1, \cdots, N\}$ **do**
4:          **for** $t_j \in id(\boldsymbol{x}^{(i)}) = \left[t_0, t_1, \cdots, t_{|\boldsymbol{x}|}\right]$ **do**
5:              **if** Rand() $< \lambda$ **then**
6:                 $t_j \leftarrow$ [MASK]          ▷ Mask tokens with the percentage $\lambda$.
7:              **else**
8:                 $t_j \leftarrow t_j$
9:              **end if**
10:          **end for**
11:          $\{t_j^{+}\} \leftarrow \text{MLM}(id(\boldsymbol{x}^{(i)}))$         ▷ Fill the mask tokens with MLM.
12:          **for** $t_j^{+} \in \{t_j^{+}\}$ **do**
13:              $emb(t_j) \leftarrow \mathbf{E}(t_j)$         ▷ Extract the embedding of the original token.
14:              $emb(t_j)^{+} \leftarrow \mathbf{E}(t_j^{+})$        ▷ Extract the embedding of the paraphrased token.
15:              $\Delta emb(t_j) \leftarrow emb(t_j)^{+} - emb(t_j)$      ▷ Measure the paraphrasing noise in the embedding domain.
16:              $emb(t_j)^{-} \leftarrow emb(t_j) - \Delta emb(t_j)$      ▷ Generate symmetrical embedding.
17:              $t_j^{-} \leftarrow \textbf{SearchNearestToken}(emb(t_j)^{-}, \mathbf{E})$
18:          **end for**
19:          $id(\boldsymbol{x}^{(i)})^{+} \leftarrow \textbf{FillMaskToken}(\{t_j^{+}\})$
20:          $id(\boldsymbol{x}^{(i)})^{-} \leftarrow \textbf{FillMaskToken}(\{t_j^{-}\})$
21:          $\boldsymbol{x}^{(i)+} \leftarrow tokenizer.decode(id(\boldsymbol{x}^{(i)})^{+})$
22:          $\boldsymbol{x}^{(i)-} \leftarrow tokenizer.decode(id(\boldsymbol{x}^{(i)})^{-})$
23:          **return** $\boldsymbol{x}^{(i)\pm}$
24:      **end for**
25: **end for**

---

Table 6: The MIA performance of SPV-MIA while applied different paraphrasing methods.

| Paraphrasing | Embedding | Semantic | Neighbour Comparing |
|---|---|---|---|
| Wiki | 0.965 | 0.951 | 0.934 |
| AG News | 0.926 | 0.903 | 0.893 |
| Xsum | 0.949 | 0.937 | 0.928 |
| Avg. | 0.944 | 0.930 | 0.918 |

## A.4 Rethinking of the Neighbour Attack

In this work, we introduce a symmetrical paraphrasing method for assessing probabilistic variation, which is motivated by a rigorous principle: detect the memorization phenomenon rather than overfitting. Meanwhile, we found that the Neighbour comparing [41] has a similar form to our proposed probabilistic variation. Thus, we further consider reformulating neighbour attack based on our intuition, then provide another explanation and motivation for it. As shown in Eq. 10, the assessment of probabilistic variation requires a pair of symmetrical paraphrased text, thus we elaborately design two paraphrasing models on embedding and semantic domains. However, it is still non-trivial to define two neighboring samples with opposite paraphrasing directions for $x$, we therefore consider directly ignoring the requirement for symmetry in the probabilistic variation. Thus, we simplify $\widetilde{x}_n^{\pm}$ to be uniformly represented by $\widetilde{x}_n$. Then we can reformulate Eq. 10 to Neighbour comparing:

$$\widetilde{p}_\theta\left(\boldsymbol{x}\right) \approx \frac{1}{2N} \sum_n^N \left(p_\theta\left(\widetilde{\boldsymbol{x}}_n^+\right) + p_\theta\left(\widetilde{\boldsymbol{x}}_n^-\right)\right) - p_\theta\left(\boldsymbol{x}\right) = \frac{1}{2N} \sum_n^{2N} p_\theta\left(\widetilde{\boldsymbol{x}}_n\right) - p_\theta\left(\boldsymbol{x}\right).  \quad (11)$$

Therefore, we believe that the neighbour attack and our proposed probabilistic variation can share the same design motivation, namely, detecting special signals that indicate the LLM has memorized training set samples. Additionally, we compared the neighbour attack with our proposed symmetric paraphrasing methods. As shown in Table 6, paraphrasing in the embedding domain achieves considerable performance gains, while paraphrasing in the semantic domain yields a marginal advantage.

## A.5 Supplementary Experimental Results

### A.5.1 Ablation Study

Table 7: Results of Ablation Study on GPT-J and LLaMA across three datasets.

| Target Model | Wiki | | AG News | | XSum | |
|---|---|---|---|---|---|---|
| | GPT-J | LLaMA | GPT-J | LLaMA | GPT-J | LLaMA |
| w/o PDC | 0.648 | 0.653 | 0.632 | 0.641 | 0.653 | 0.661 |
| w/o PVA | 0.901 | 0.913 | 0.864 | 0.885 | 0.873 | 0.919 |
| SPV-MIA | 0.929 | 0.951 | 0.885 | 0.903 | 0.897 | 0.937 |

In the previous experiments, we have validated the superiority of our proposed SPV-MIA over existing algorithms, as well as its versatility in addressing various challenging scenarios. However, the specific contributions proposed by each module we proposed are still unknown. In this subsection, we conduct an ablation study to audit the performance gain provided by the two proposed modules. Concretely, we respectively remove the practical difficulty calibration (PDC) and probabilistic variation assessment (PVA) that we introduced in Section 4.2 and Section 4.3. The results are represented in Table 7, where each module contributes a certain improvement to our proposed method. Besides, the PVC approach seems to play a more critical role, which can still serve as a valid adversary without the PVA. Thus, in practical scenarios, we can consider removing the PVA to reduce the frequency of accessing public APIs.

### A.5.2 TPR@0.1%FPR

As a supplement to the TPR@1%FPR results represented in Table 2, we further provide the performance measured by TPR@0.1%FPR in Table 8.

| Method | Wiki | | | | | AG News | | | | | Xsum | | | | |
|---|---|---|---|---|---|---|---|---|---|---|---|---|---|---|---|
| | GPT-2 | GPT-J | Falcon | LLaMA | Avg. | GPT-2 | GPT-J | Falcon | LLaMA | Avg. | GPT-2 | GPT-J | Falcon | LLaMA | Avg. |
| Loss Attack | 0.1% | 0.1% | 0.1% | 0.1% | 0.1% | 0.1% | 0.1% | 0.1% | 0.2% | 0.1% | 0.1% | 0.2% | 0.1% | 0.1% | 0.1% |
| Neighbour Attack | 1.2% | 0.6% | 0.5% | 0.8% | 0.8% | 0.8% | 0.4% | 0.3% | 0.4% | 0.5% | 0.5% | 0.3% | 0.3% | 0.3% | 0.4% |
| DetectGPT | 0.9% | 0.4% | 0.5% | 0.6% | 0.6% | 0.6% | 0.2% | 0.3% | 0.4% | 0.4% | 0.4% | 0.2% | 0.2% | 0.3% | 0.3% |
| Min-K% | 1.4% | 0.6% | 0.7% | 0.9% | 0.9% | 1.2% | 0.4% | 0.7% | 0.6% | 0.7% | 0.5% | 0.3% | 0.4% | 0.4% | 0.4% |
| Min-K%++ | 1.2% | 0.7% | 0.9% | 1.1% | 1.0% | 1.4% | 0.9% | 1.1% | 1.0% | 1.1% | 0.8% | 0.4% | 0.7% | 0.6% | 0.6% |
| LiRA-Base | 1.9% | 1.4% | 1.5% | 1.7% | 1.6% | 1.8% | 1.4% | 1.3% | 1.4% | 1.5% | 3.1% | 2.5% | 2.6% | 3.5% | 2.9% |
| LiRA-Candidate | 3.7% | 2.5% | 2.8% | 3.2% | 3.1% | 2.3% | 1.8% | 1.7% | 1.9% | 1.9% | 4.7% | 3.4% | 3.8% | 5.1% | 4.3% |
| SPV-MIA | **39.1%** | **28.9%** | **32.7%** | **37.8%** | **34.6%** | **25.3%** | **17.3%** | **18.7%** | **23.5%** | **21.2%** | **34.4%** | **27.6%** | **28.9%** | **31.5%** | **30.6%** |

Table 8: **TPR@0.1%FPR** for detecting member texts from four LLMs across three datasets for SPV-MIA and five previously proposed methods. **Bold** and Underline respectively represent the best and the second-best results within each column (model-dataset pair).

### A.5.3 AUC Curves

As a supplement to the main experimental results represented in Table 1, we further provide the raw ROC curve for a more comprehensive presentation in Fig. 6 (linear-scale) and Fig. 7 (log-scale).

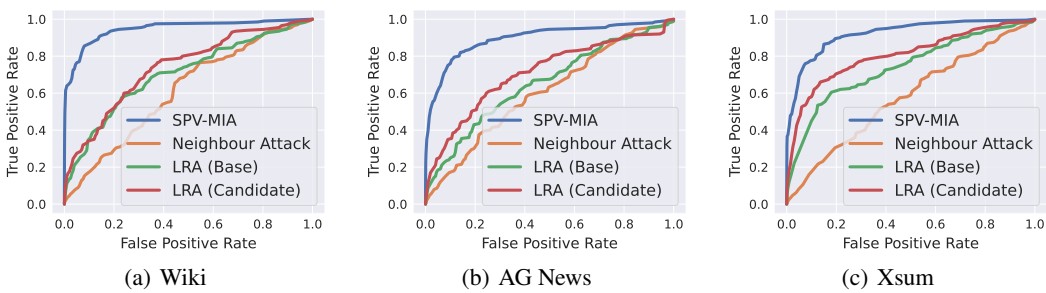

(a) Wiki  (b) AG News  (c) Xsum

Figure 6: **Linear-scale** ROC curves of SPV-MIA and the top-three best baselines on LLaMAs fine-tuned over three datasets.

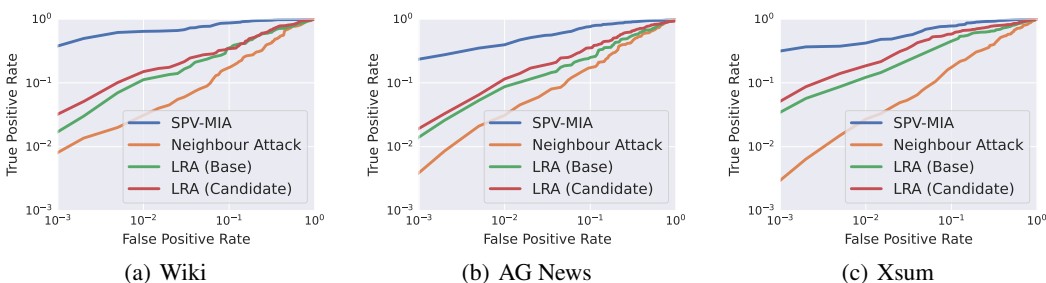

(a) Wiki  (b) AG News  (c) Xsum

Figure 7: **Log-scale ROC curves** of SPV-MIA and the three representative baselines on LLaMAs fine-tuned over three datasets.

### A.5.4 Performance of Target LLMs

We supplemented the performance of all LLM-dataset pairs on both the training and test sets (estimated using PPL). As shown in Table 9, the experimental results indicate that none of the fine-tuned LLMs exhibit significant overfitting, which aligns with our claim in the main body. Additionally, we provided the performance of the LLM under different privacy budgets $\epsilon$, as shown in Table 10.

### A.5.5 Performance of Representative Baselines under Different Privacy Budgets

We have conducted three representative baseline attacks for the DP-SGD model (LLaMA fine-tuned over Ag News dataset) and compared it with our method (SPV-MIA). The results are provided in Table 11. The results demonstrate that SPV-MIA consistently maintains substantial MIA performance margins over different settings of the privacy budget.

Table 9: The perplexity (PPL) of each LLM-dataset pair on training set and test set.

| Target Model | GPT-2 | | GPT-J | | Falcon | | LLaMA | |
|---|---|---|---|---|---|---|---|---|
| | Training | Test | Training | Test | Training | Test | Training | Test |
| Wiki | 25.34 | 27.47 | 12.35 | 12.44 | 7.26 | 7.73 | 7.00 | 7.43 |
| AG News | 23.86 | 26.34 | 12.49 | 13.24 | 8.92 | 9.54 | 9.03 | 9.13 |
| Xsum | 26.42 | 28.31 | 13.43 | 13.88 | 7.69 | 7.96 | 7.35 | 7.65 |

Table 10: The perplexity (PPL) of each LLM-dataset pair trained w.r.t different privacy budget $\epsilon$.

| Privacy Budget $\epsilon$ | 15 | 30 | 60 | + inf |
|---|---|---|---|---|
| Wiki | 8.45 | 8.16 | 7.76 | 7.43 |
| AG News | 10.84 | 9.68 | 9.32 | 9.13 |
| Xsum | 8.89 | 8.33 | 7.98 | 7.65 |

## A.6 Experimental Settings

In this subsection, we give a extensive introduction of experimental settings, including the datasets, target LLMs and baselines, as well as the implementation details.

### A.6.1 Datasets

Our experiments utilize six different datasets across multiple domains and LLM use cases, where we employ three datasets as the private datasets to fine-tune the target LLMs, and the remaining datasets as the public datasets from the exact domains. Specifically, we use the representative articles on Wikitext-103 dataset [43] to represent academic writing tasks, news topics from the AG News dataset [78] to represent news topic discussion task, and documents from the XSum dataset [49] to represent the article writing task. Besides, we utilize Wikicorpus [55], TLDR News [30], and CNNDM [23] datasets to respectively represent as the publicly accessible dataset from the same domain for each task.

### A.6.2 Target Large Language Models

To obtain a comprehensive evaluation result, we conduct our experiments over four well-known and widely adopted LLMs as the pre-trained models with different scales from 1.5B parameters to 7B parameters:

- **GPT-2 [54]:** It is a transformer-based language model released by OpenAI in 2019, which has 1.5 billion parameters and is capable of generating high-quality text samples.

- **GPT-J [69]:** It is an open-source LLM released by EleutherAI in 2021 as a variant of GPT-3. GPT-J has 6 billion parameters and is designed to generate human-like with appropriate prompts.

- **Falcon-7B [3]:** Falcon is a family of state-of-the-art LLMs created by the Technology Innovation Institute in 2023. Falcon has 40 billion parameters, and Falcon-7B is the smaller version with less consumption.

- **LLaMA-7B [65]:** LLaMA is one of the most state-of-the-art LLM family open-sourced by Meta AI in 2023, which has outperformed other open-source LLMs on various NLP benchmarks. It has 65 billion parameters and has the potential to accomplish advanced tasks, such as code generation. In this work, we utilize the lightweight version, LLaMA-7B.

### A.6.3 Baselines

We choose six MIAs designed for LMs to comprehensively evaluate our proposed method, including three reference-free attacks and one reference-based attack with one variant.

- **Loss Attack [73]:** A standard metric-based MIA that distinguishes member records simply by judging whether their losses are above a preset threshold.

- **Neighbour Attack [41]:** The Neighbour Attack avoids using a reference model to calibrate the loss scores and instead utilizes the average loss of plausible neighbor texts as the benchmark.

Table 11: The AUC performance of SPV-MIA and three representative baselines w.r.t different privacy budget $\epsilon$.

| Privacy Budget $\epsilon$ | 15 | 30 | 60 | $+\inf$ |
|---|---|---|---|---|
| Loss Attack | 0.523 | 0.551 | 0.568 | 0.580 |
| Neighbour Attack | 0.542 | 0.564 | 0.587 | 0.610 |
| LIRA-Candidate | 0.611 | 0.655 | 0.684 | 0.714 |
| SPV-MIA | 0.766 | 0.814 | 0.852 | 0.903 |

- **DetectGPT [48]:** A zero-shot machine-generated text detection method. Although DetectGPT is specially designed for LLMs-generated text detection, but has the potential to be adapted for identifying the text utilized for model training.

- **Min-K% [59]** An MIA method designed for pre-trained LLMs, which evaluate the token-level probability and employ the average over the $k\%$ lowest probability as the MIA metric.

- **Min-K%++ [76]** An enhanced version of Min-K% that utilizes a more sophisticated mechanism to detect the records with relatively high probability curvature.

- **Likelihood Ratio Attack (LiRA-Base) [47]:** A reference-based attack, which adopts the pre-trained model as the reference model to calibrate the likelihood metric to infer membership.

- **LiRA-Candidate [47]:** A variant version of LiRA, which utilizes a publicly available dataset in the same domain as the training set to fine-tune the reference model.

### A.6.4 Detailed Information for Reproduction

Table 12: Detailed split and other information of datasets.

| Dataset | Relative Datasets | | Target Model | | Reference Model | |
|---|---|---|---|---|---|---|
| | Domain-specific | Irrelevant | # Member | # Non-member | # Member | # Non-member |
| Wikitext-103 | Wikicorpus | AG News | 10,000 | 1,000 | 10,000 | 1,000 |
| AG News | TLDR News | Xsum | 10,000 | 1,000 | 10,000 | 1,000 |
| Xsum | CNNDM | Wikitext-103 | 10,000 | 1,000 | 10,000 | 1,000 |

All experiments are compiled and tested on a Linux server (CPU: AMD EPYC-7763, GPU: NVIDIA GeForce RTX 3090), Each set of experiments for the LLM-dataset pairs took approximately 8 hours, and we spent around 14 days completing all the experiments. For each dataset, we pack multiple tokenized sequences into a single input, which can effectively reduce computational consumption without sacrificing performance [33]. Besides, the packing length is set to 128 tokens. Then, we use 10,000 samples for fine-tuning over pre-trained LLMs and 1,000 samples for evaluation. The detailed information of datasets is summarized in Table 12. For each target LLM, we let it fine-tuned with the training batch size of 16, and trained for 10 epochs. The learning rate is set to 0.0001. We adopt the AdamW optimizer [40] to achieve the generalization of LLMs, which is composed of the Adam optimizer [32] and the L2 regularization. For GPT-2, which has a relatively small scale, we adopt the full fine-tuning, which means all parameters are trainable. For other LLMs that are larger, we utilize a parameter-efficient fine-tuning method, Low-Rank Adaptation (LoRA) [24], as the default fine-tuning method. For the paraphrasing model in the embedding domain, the Gaussian noise scale is set to $\sigma = 0.05$. For the paraphrasing model in the semantic domain, the paraphrasing percentage is set to $\lambda = 0.2$. For both of the two paraphrasing models, we generate 10 symmetrical paraphrased text pairs for each target text record. For the reference LLM fine-tuned with our proposed self-prompt approach, we utilize the domain-specific data as the default prompt text source. Then, we collect 10,000 generated texts from target LLMs with an equal length of 128 tokens to construct reference datasets. We fine-tune the reference LLM for 4 epochs and the training batch size of 16.

