# OpenReview forum: "Membership Inference Attacks against Fine-tuned Large Language Models via Self-prompt Calibration"
_NeurIPS.cc/2024/Conference — NeurIPS 2024 poster_

### Official Review · Reviewer_i2LA · 2024-07-12

**Soundness:** 3
**Presentation:** 1
**Contribution:** 2
**Rating:** 7
**Confidence:** 5

**Summary:**

In this work, the authors propose an MIA for LLMs that utilizes prompt calibration to measure variation on model behavior for neighboring inputs. The authors also make a connection with the neighborhood attack from Mattern et al and show how their framework encapsulates such neighborhood-based attacks. Performance on multiple domains demonstrate a clear bump in performance using this attack.

**Strengths:**

- Attempts to improve black-box privacy auditing are useful and needed, especially with more and more model trainers where access is available only via wrappers or APIs.
- Figures and diagrams in the paper are very well made and helpful, and complement the writing well.
- Assessing the method's robustness for different sources of prompts is useful and helps understand worst-case performance when prompt data is completely unrelated to the member distribution.

**Weaknesses:**

- The core idea here is not very different from distillation-based stealing [1], followed by a slightly modified neighborhood attack.

- L240: There is no clear relationship between "modest" paraphrasing in token space, and a corresponding $h$ in the Equation 10, let alone plus/minus. For a language model, $x$ is in the embedding space, so small perturbations to the embeddings will probably not correspond to any actual tokens. Moreover, inputs are sequences and thus perturbations to multiple tokens makes the assumption around a small $h$ even less plausible. Replacing 20% of all tokens is in no way a "modest" replacement.

- In Table 1: why not include a version in the baseline that uses the reference models that you train but does not use the neighborhood logic. This would help better understand the contribution of both these steps, serving as an ablation study.

- L631-633: This is not a grounded statement. Making the connection not only seems unnecessary, but incorrect. While it is nice to have theoretical connections, it is by no means necessary for a good paper. I would urge the authors to rethink the use of theory to make connections in places where they do not seem valid without handwavy justifications.

## Minor comments
- L32: '[44]' is merely a presentation abstract. [2], for instance, provides an actual systemization of memorization in LLMs
- Figure 1b - what is "memorization"? Figures should be labeled clearly.
- The authors seem to be confused about which works introduced LiRA. The authors seem to suggest that Mireshghallah et al introduced LiRA (L45, L102), whereas it was proposed by Carlini et al. [3].
- L49-50: While overfitting means higher leakage, it is by no means an "assumption" for MIAs.
- L59: "Exponential" is a bold claim here; there are only 3 datapoints. It is not surprising that it declines, as it is explained by theory [4]
- L81: Computing a second-order derivative in an LLM is just not feasible practically.
- L84: The neighborhood-based MIA requires another masking model, which need not be an LLM.
- L163: If it is over-represented in dataset, it is needless to say a member. The statement here feels vacuous.
- L164: Please also cite appropriate work that introduces this formal dependency on reference models for difficulty calibration [4].
- L168: Missing citation
- The dot on top of $\theta$ in Equation 5 (and other related equations) is barely visible- please use better notation.
- L182: Missing citation
- L229: Why is a negative-definite Hessian problematic? In practice, the (empirical) Hessian is very low-rank and most of its entries are close to zero.
- L235: "..can be interpreted as kind of" - this is very handwavy
- Table 1: Please also include TPR@FPRs (like 1% and 0.1% FPR). The table is also missing a lot of attacks like Min-K%++ [5] and MoPE [6]
- Figure 3 Why is Y axis going above AUC = 1.0?


### References
- [1] Galichin, Andrey V., et al. "GLiRA: Black-Box Membership Inference Attack via Knowledge Distillation." arXiv preprint arXiv:2405.07562 (2024).
- [2] Hartmann, Valentin, et al. "Sok: Memorization in general-purpose large language models." arXiv preprint arXiv:2310.18362 (2023).
- [3] Carlini, Nicholas, et al. "Membership inference attacks from first principles." 2022 IEEE Symposium on Security and Privacy (SP). IEEE, 2022.
- [4] Sablayrolles, Alexandre, et al. "White-box vs black-box: Bayes optimal strategies for membership inference." International Conference on Machine Learning. PMLR, 2019.
- [5] Zhang, Jingyang, et al. "Min-K%++: Improved Baseline for Detecting Pre-Training Data from Large Language Models." arXiv preprint arXiv:2404.02936 (2024).
- [6] Li, Marvin, et al. "Mope: Model perturbation-based privacy attacks on language models." arXiv preprint arXiv:2310.14369 (2023).

**Questions:**

- In the codebase, at line 65 of `ft_llms/refer_data_generate.py`, the prompt data is sampled directly from 10K-20K range of samples from the train data, so there is a very clear bias here (supposed to be from same "distribution"?) If prompt data is indeed sampled from train data, completions would likely be the actual data itself, so reference model would behave similar to target model. Scores would thus be $m(x) - m(\hat{x}) \approx 0$ for train data, and non-zero for validation data. If this is indeed true, this is a **big empirical flaw**. I would urge the authors to clarify, and also plot the distributions of actual scores corresponding to $m(x)$. I may consider increasing my score if the authors can look into.

- Why is there no mention of any studies related to membership inference for LLMs in Section 2 (Large Language Models)? There have been plenty specifically looking at LLMs [1,2,3,4].

- L147: "adversary has no prior information" - Does it have access to the model's initial weights?

- Figure 5b: Why is there a drop in performance for the domain-specific case?

### References
- [1] Maini, Pratyush, et al. "LLM Dataset Inference: Did you train on my dataset?." arXiv preprint arXiv:2406.06443 (2024).
- [2] Mozaffari, Hamid, and Virendra J. Marathe. "Semantic Membership Inference Attack against Large Language Models." arXiv preprint arXiv:2406.10218 (2024).
- [3] Meeus, Matthieu, et al. "Inherent Challenges of Post-Hoc Membership Inference for Large Language Models." arXiv preprint arXiv:2406.17975 (2024).
- [4] Duan, Michael, et al. "Do membership inference attacks work on large language models?." arXiv preprint arXiv:2402.07841 (2024).
- [5] Das, Debeshee, Jie Zhang, and Florian Tramèr. "Blind Baselines Beat Membership Inference Attacks for Foundation Models." arXiv preprint arXiv:2406.16201 (2024).

**Limitations:**

There is no proper discussion about limitations, apart from a 1-2 sentences in the Appendix.

---

> ### Author Rebuttal · Authors · 2024-08-07
>
> Dear reviewer i2LA,
>
> We deeply appreciate the time you took to review our work and your meticulous comments for improvement. Below we address the questions raised in your review, the responses to the weaknesses and minor comments, as well as the reference can be found in subsequent comments:
>
> **Q1: In the codebase, is the prompt data sampled directly from the fine-tuning data? If this is indeed true, this is a big empirical flaw.**
> We want to clarify that we did not directly extract prompt data from the fine-tuning data, this is simply a part of the code that is easily misunderstood. Specifically, the `train_dataset` variable in line 65 of the `ft_llms/refer_data_generate.py` file is not the actual training set used to fine-tune the target model. We chose to extract prompt data from the `train_dataset` variable because, in some datasets, the number of samples in the `valid_dataset` is small and may lead to out-of-range errors. In fact, when preparing the fine-tuning dataset for the target model, we will manually skip the 10k-20k range: see line 226 in the `ft_llms/llms_finetune.py` file:
>
> `226: train_dataset = Dataset.from_dict(train_dataset[args.train_sta_idx:args.train_end_idx])`
>
> Moreover, we have experimentally demonstrated that the source of the prompt data has almost no impact on the performance of our proposed method (In Section 5.4, Fig. 4). Even prompt data from arbitrary unrelated sources can achieve excellent performance.
>
> Furthermore, we are committed to fully addressing your concern by adhering to your guidance to plot the distributions of metric score $\Delta m(x)$ (in the PDF file of the global comment, Figure 2), for the training data (member) and validation data (non-member). The results clearly demonstrate that $\Delta m(x) = m(x) - \widetilde{m}(x)$ is not approximate to zero for the training data. Additionally, the training data metric score is generally higher than the validation data, indicating that the reference model would behave very differently from the target model regarding the target model's training data. All these results demonstrate that we did not directly extract prompt data from the fine-tuning data.
>
>
> **Q2: Why is there no mention of any studies related to membership inference for LLMs in Section 2 (Large Language Models)? There have been plenty specifically looking at LLMs [1,2,3,4,5]**
>
> We have reviewed the current membership inference attacks on LLMs in Section 2 (Membership Inference Attack) and therefore did not replicate this content in Section 2 (Large Language Models). Moreover, since most of the literature [1,2,3,5] you recommended was submitted to arXiv in June this year (after the NeurIPS deadline), we are not able to discuss these papers in our submitted version. Thank you for the reminder, we will add a comprehensive discussion of the literature you recommended in Section 2 (Membership Inference Attack).
>
>
>
> **Q3: L147: "adversary has no prior information" - Does it have access to the model's initial weights?**
>
> > Explanation for L147: "adversary has no prior information about which data records are utilized for fine-tuning."
>
> On line 147, we emphasize that our method does not have prior information about the fine-tuning dataset of the target model. Existing reference-based methods heavily rely on the strong assumption that the adversary can obtain a dataset that is from the same distribution but does not intersect with the fine-tuning dataset. Our proposed self-prompt method avoids this assumption.
>
> > Answer for the question: Does it have access to the model's initial weights?
>
> The proposed method does not necessarily require the initial parameters of the model. An API that allows the adversary to upload a customized dataset for fine-tuning would also meet the conditions. Therefore, our proposed method is not only applicable to open-source models with publicly available parameters but also for closed-source models that provide fine-tuning APIs (such as ChatGPT [6]). Thank you for your question for improvement, we will include this discussion in the camera-ready version.
>
> Furthermore, to further allay your concerns, we consider a more strict scenario where there is no access (even the fine-tuning API) to the pre-trained model corresponding to the target model. We employ LLaMA-7 fine-tuned over the Agnew dataset as the target model, and let the adversary fine-tune over different pre-trained models (GPT-J, Falcon and LLaMA are adopted) to prepare the reference model. The results are shown in the table below. From the experimental results in the table, we can find that the proposed method still maintains a large performance margin over the baselines that can fine-tune the pre-trained model of the target model. We will include these results in the camera-ready version.
>
> |Metric|AUC|TPR@1%FPR|TPR@0.1\%FPR|
> |----------------------|-----|---------|-----------|
> |SPV-MIA (GPT-J)|0.832|19.3\%|7.5\%|
> |SPV-MIA (Falcon)|0.812|21.7\%|8.7\%|
> |SPV-MIA |0.903|39.5\%|23.5\%|
> |LiRA-Base |0.657|8.7\%|1.4\%|
> |LiRA-Candidate |0.714|11.5\%|1.9\%|
>
> **Q4: Figure 5b: Why is there a drop in performance for the domain-specific case?**
>
> This phenomenon occurs because the performance of reference-based MIA is highly dependent on the quality (source) of the reference dataset (in Section 5.3, Figure 3). Although a domain-specific dataset pertains to the same domain as the target model's training set, there can still be significant distributional differences, even if these differences are smaller compared to an irrelevant dataset. Moreover, we have conducted experiments that demonstrate the quality of the domain-specific dataset is relatively low (in Section 5.3, Figure 3). Therefore, using excessively long prompt text lengths will cause the reference dataset extracted from the target model via the self-prompt method to be more similar in data distribution to the domain-specific dataset (low quality) rather than the target model's training dataset (high quality).

---

> ### Author Response · Authors · 2024-08-07
> **(Optional for reviewer to read) The responses to the weaknesses -- Part 1**
>
> To answer the detailed questions of i2LA, we provide more information in the form of comments. We understand reviewers are not obligated to read comments. we keep them just in case i2LA are interested.
>
> **W1: The core idea here is not very different from distillation-based stealing [7], followed by a slightly modified neighborhood attack.**
>
> > **W1.1: The core idea here is not very different from distillation-based stealing (GLiRA) [7]**
>
> GLiRA [7] is a related work that also focuses on MIA. Although our ideas are slightly similar, the idea of SPV-MIA still originated independently and was proposed earlier. A direct piece of evidence is that GLiRA is a recent publication, submitted on arXiv shortly before the NeurIPS ddl (May 13). Moreover, our proposed method significantly diverges from it in multiple aspects:
>
> 1. **The Target Model**:
>  GLiRA is an MIA method specifically tailored for classification models. Thus, certain specific designs are only applicable to classification models and cannot be applied to LLMs we focus on. For example, GLiRA requires training a large number of shadow models (128 in their experiments), a level of overhead that is not negligible when the target model is an LLM.
> 2. **The Adversary Knowledge**:
>  GLiRA assumes that the adversary can directly sample data points from the same data distribution as the training dataset of the target model. This is a widely acknowledged unrealistic assumption, especially in the context of LLMs [8,9,10], as their training datasets are typically not publicly available. In contrast, our proposed method has no prior knowledge about the data distribution utilized for fine-tuning. We have avoided this assumption with the proposed self-prompt method.
> 3. **The Essential Motivation**:
>  Although GLiRA and the self-prompt method we proposed both share the commonality of mimicking the target model, the motivation driving each approach is different. Specifically, GLiRA essentially relies on hypothesis testing with a large number of shadow models, and they hypothesize that using knowledge distillation to mimic the target model can significantly improve the accuracy of MIA. However, the self-prompt method we proposed is to fine-tune a high-quality reference model without sampling data from the same training data distribution as the target model.
>
> Therefore, it is difficult to say that our proposed method is not very different from GLiRA. Thanks for your reminder, we will discuss your suggested paper in the camera-ready version.
>
> > **W1.2: Followed by a slightly modified neighborhood attack**
>
> Our method and the neighbour attack approach differ significantly in concept, although they both employ text paraphrasing for attacking. The neighbour Attack is motivated by exploring a substitution of reference models. In this work, we provide another motivation based on model memorization, that is, the member text memorized by the model will tend to be located on the local maximum in the generative distribution [11]. Therefore, this study may inspire the following studies to design more sophisticated methods for inferring membership by characterizing the model memorization. For example, a recently published work (released after the NeurIPS deadline) that you also recommended [9] proposed an MIA against pre-trained LLMs based on the concept of local maximum detection.
>
> **W2: L240: There is no clear relationship between "modest" paraphrasing in token space, and a corresponding $h$ in the Equation 10, let alone plus/minus. For a language model, is in the embedding space, so small perturbations to the embeddings will probably not correspond to any actual tokens. Moreover, inputs are sequences and thus perturbations to multiple tokens makes the assumption around a small even less plausible. Replacing 20% of all tokens is in no way a "modest" replacement.**
>
> Equation 10 is approximately derived from Equation 9, requiring $h \to 0$, which corresponds to conducting modest perturbations in the high-dimensional semantic space of sentence $\boldsymbol{x}$. Thus, the perturbation is conducted at the sentence-level rather than the token-level. Since the mask-filling model samples sentences similar to $\boldsymbol{x}$ with minimal changes to semantic meaning, we can think of the mask-filling model as first sampling a similar semantic embedding $\boldsymbol{\widetilde{e}}$ and then mapping this to a partial paraphrased token sequence ($\boldsymbol{\widetilde{e}}$ → $\boldsymbol{\widetilde{x}}$). Thus, the modest perturbation in the sentence space can lead to partial tokens in this sentence be paraphrased. We chose a 20% perturbation rate over a smaller one because a reduced perturbation rate would render the numerical variances between the original and paraphrased sentences imperceptible in the score metrics. This scenario could lead to the metrics we intend to assess having numerically insignificant values, potentially overshadowed by the noise introduced through numerical computations.

---

> ### Author Response · Authors · 2024-08-07
> **(Optional for reviewer to read) The responses to the weaknesses -- Part 2**
>
> To answer the detailed questions of i2LA, we provide more information in the form of comments. We understand reviewers are not obligated to read comments. we keep them just in case i2LA are interested.
>
> **W3: In Table 1: why not include a version in the baseline that uses the reference models that you train but does not use the neighborhood logic. This would help better understand the contribution of both these steps, serving as an ablation study.**
>
> We have shown the ablation study in Appendix A.5.1 for auditing the contributions of the proposed Practical Difficulty Calibration (PDC) and probabilistic variation assessment (PVA, i.e., the neighborhood logic) separately. Furthermore, we also have conducted an extensive ablation study in Appendix A.4 to investigate the contributions of our proposed two symmetrical paraphrasing methods compared with neighbour attack, in which we adopt the two symmetrical paraphrasing methods, and neighbour attack as the PVA module, respectively. Thank you for the reminder, we will add SPV-MIA w/o PVA in Table 1 in the camera-ready version.
> In terms of these ablation studies shown in the Appendix, we quote the results and merge them into one table as follows:
>
> Table: Results of Ablation Study on LLaMA across three datasets.
> |Dataset|Wiki|AG News|XSum|Avg.|
> |--------------------------------------|---------|---------|---------|---------|
> |SPV-MIA (Embedding-based Paraphrasing)|0.956|0.926|0.949|0.944|
> |SPV-MIA (Semantic-based Paraphrasing)|0.951|0.903|0.937|0.930|
> |SPV-MIA (Neighbour Attack)|0.934|0.893|0.928|0.918|
> |**w/o PVA**|**0.913**|**0.885**|**0.919**|**0.906**|
> |w/o PDC|0.653|0.641|0.661|0.652|
>
> The results demonstrate that the PDC approach seems to play a more critical role, and can still serve as a valid adversary without the PVA. Thus, in practical scenarios, we can consider removing the PVA to reduce the frequency of accessing public APIs.
>
> **W4: L631-633: This is not a grounded statement. Making the connection not only seems unnecessary, but incorrect. While it is nice to have theoretical connections, it is by no means necessary for a good paper. I would urge the authors to rethink the use of theory to make connections in places where they do not seem valid without handwavy justifications.**
>
> Lines 631-633 in Appendix "A.4 Reformulate the Neighbour Attack" only provide a new perspective to understand paraphrasing-based methods. Specifically, we want to convey that the proposed PVA and neighbour attack may have the same potential motivation, which could inspire subsequent work to identify member samples by detecting local maxima. For example, a recent paper you recommended also employs this idea [9]. Considering that the content here is not sufficient to support a compact theoretical framework, we will change the title to "A.4 Rethinking of the Neighbour Attack" and replace the declarative tone with a conjectural tone.

---

> ### Author Response · Authors · 2024-08-07
> **(Optional for reviewer to read) The responses to the minor comments**
>
> To answer the detailed questions of i2LA, we provide more information in the form of comments. We understand reviewers are not obligated to read comments. we keep them just in case i2LA are interested.
>
> **Minor Comments: For M1, M2, M3, M4, M5, M7, M9, M10, M11, M12, M14**
>
> We are very grateful for your careful review and detailed comments on our paper. We will follow your instructions to revise these minor issues in the camera-ready version.
>
> **M6: L81:Computing a second-order derivative in an LLM is just not feasible practically.**
>
> We did not directly calculate the second-order derivatives in LLMs; instead, we estimate features of partial second-order derivatives by calculating the score differences between the target text and its paraphrased versions. Similarly, the paper you recommended [9] also used the concept of estimating second-order derivatives on LLMs for membership inference and utilized statistical methods to characterize the features of the second-order derivatives. We will revise the corresponding wording in the camera-ready version.
>
> **M8: L163:If it is over-represented in dataset, it is needless to say a member. The statement here feels vacuous.**
>
> We want to clarify that an over-represented record in the data distribution is not necessarily a member. This is a widely recognized statement claimed by a highly influential paper [12] in the field of MIA. Specifically, a non-member sample may have a high membership score simply because it is over-represented in the data distribution. Consequently, an attack that determines a sample is likely to be a member due to having a high score will inevitably fail on these over-represented samples. This is the reason why difficulty calibration via the reference model is proposed.
>
> **M13: L229:Why is a negative-definite Hessian problematic? In practice, the (empirical) Hessian is very low-rank and most of its entries are close to zero.**
>
> Because existing research has shown that member samples fall at the local maximum of the probability distribution, where the Hessian matrix is negative definite [13]. Although we derived our method through the negative definiteness of the Hessian matrix, we did not directly calculate the Hessian matrix because this is impractical in the LLM scenario. Therefore, we translated it into an estimation of probability variation metrics, which can be calculated via our proposed paraphrasing method.
>
> **M15: Table 1: Please also include TPR@FPRs (like 1% and 0.1% FPR). The table is also missing a lot of attacks like Min-K%++ [9] and MoPE [2]**
>
> > M15.1: Table 1: Please also include TPR@FPRs (like 1% and 0.1% FPR).
>
> We have followed your suggestion and added two new evaluation metrics: TPR@1%FPR, and TPR@0.1%FPR. All additional results can be found in the PDF file of the global response. Thank you for your valuable suggestion for improvement. We will update these results to the camera-ready version. We quote partial results in the tables below. Experimental results show that, compared to AUC, SPV-MIA achieves a larger performance margin under new metrics. This indicates that, at lower FPRs, SPV-MIA maintains a higher TPR compared to the baselines.
>
> > The table is also missing a lot of attacks like Min-K%++ [9] and MoPE [14]
>
> We have added Min-K\%++ [9] as a baseline based on your suggestion. Since MoPE [14] does not provide open-source code, we have attempted to contact the authors to obtain the code and have committed to supplementing the relevant experiments upon receiving it. As an alternative, we suggest adding Min-K\% [8] as a baseline, as it can achieve better performance in some cases.
>
> The supplementary experiments are shown in the PDF file of the global response. In the experimental results across 3 datasets and 4 LLMs, Min-K\%  and Min-K\%++ achieve the best or second-best performance among 5 reference-free baselines. We will update these results in the camera-ready
> version. We quote the partial results in the table below.
>
> Table. Evaluation of all baselines and SPV-MIA on LLaMA@AG News Dataset using TPR@1\%FPR and TPR@0.1\%FPR
> |Methods|Loss Attack|Neighbour Attack|DetectGPT|Min-K\%|Min-K\%++|LiRA-Base|LiRA-Candidate|SPV-MIA|
> |------------|-----------|----------------|---------|-------|---------|---------|--------------|-------|
> |AUC|0.580|0.610|0.603|0.619|0.631|0.657|0.714|0.903|
> |TPR@1\%FPR|1.2\%|3.1\%|2.7\%|3.6\%|4.1\%|8.7\%|11.5\%|39.5\%|
> |TPR@0.1\%FPR|0.2\%|0.4\%|0.4\%|0.6\%|1.0\%|1.4\%|1.9\%|23.5\%|
>
> **M17 Figure 3 Why is Y axis going above AUC = 1.0?**
>
> The values of the data points within the histogram in Fig 3 do not exceed AUC=1. The y-axis range was drawn larger to accommodate the legend and to make the figure clearer.

---

> ### Author Response · Authors · 2024-08-07
> **(Optional for reviewer to read) The reference of rebuttal**
>
> # Reference
>
> [1] Maini, et al. "LLM Dataset Inference: Did you train on my dataset?" arXiv 2024.
>
> [2] Mozaffari, Marathe. "Semantic Membership Inference Attack against Large Language Models." arXiv 2024.
>
> [3] Meeus, et al. "Inherent Challenges of Post-Hoc Membership Inference for Large Language Models." arXiv 2024.
>
> [4] Duan, et al. "Do membership inference attacks work on large language models?" arXiv 2024.
>
> [5] Das, Zhang, Tramèr. "Blind Baselines Beat Membership Inference Attacks for Foundation Models." arXiv 2024.
>
> [6] Peng, et al. "Gpt-3.5 turbo finetuning and api updates." 2023.
>
> [7] Galichin, et al. "GLiRA: Black-Box Membership Inference Attack via Knowledge Distillation." arXiv 2024.
>
> [8] Shi, et al. "Detecting Pretraining Data from Large Language Models." ICLR 2024.
>
> [9] Zhang, et al. "Min-K%++: Improved Baseline for Detecting Pre-Training Data from Large Language Models." arXiv 2024.
>
> [10] Mattern, et al. "Membership Inference Attacks against Language Models via Neighbourhood Comparison." ACL 2023.
>
> [11] van den Burg, Williams. "On memorization in probabilistic deep generative models." NeurIPS 2021.
>
> [12] Watson, et al. "On the Importance of Difficulty Calibration in Membership Inference Attacks." ICLR 2021.
>
> [13] Boyd, Vandenberghe. "Convex optimization." CUP 2004.

---

> > ### Comment · Reviewer_i2LA · 2024-08-11
> >
> > Thank you to the authors for their responses. I appreciate all the effort in responding to my questions and concerns so thoroughly. I do not have any more questions, and am increasing the score to 7. Good luck!

---

> > > ### Author Response · Authors · 2024-08-14
> > >
> > > Dear Reviewer i2LA,
> > >
> > > Thank you for taking the time to review our responses and for incresing the score. We are glad that our responses addressed all your questions and concerns. Thank you once again for all your efforts.
> > >
> > > Best regards,
> > >
> > > The Authors

---

### Official Review · Reviewer_saYA · 2024-07-12

**Soundness:** 2
**Presentation:** 3
**Contribution:** 2
**Rating:** 5
**Confidence:** 4

**Summary:**

This paper presents a membership inference attack against causal language models, addressing the limitations of previous attacks, such as the inaccessibility of appropriate reference datasets and heavy reliance on overfitting. To overcome these limitations, the authors propose a self-prompt approach to extract reference datasets from LLMs and introduce a membership signal based on memorization. They compare their method with other methods using the AUC metric.

**Strengths:**

- The paper effectively identifies two key limitations of previous membership inference attacks against LLMs, providing a clear motivation for the proposed targeted solutions.

- The proposed probabilistic variation assessment is interesting and has potential applications beyond LLMs.

- The paper is well-written with a clear logical structure, making it easy to follow.

**Weaknesses:**

- The main evaluation metrics need to be revised. As established by Carlini et al. [R1], membership inference attacks should be evaluated by computing their true-positive rate at low (e.g., ≤ 0.1%) false-positive rates, rather than using average-case metrics like AUC. Such evaluation metrics have become the de facto standard in evaluating membership inference attacks and are used by many existing works [R2] [R3]. I suggest using two metrics: the Full Log-scale Receiver Operating Characteristic (ROC) Curve to highlight low false-positive rates, and the TPR at a low FPR, which measures attack performance at specific FPRs (e.g., 0.1%, 0.01%).

- The experimental settings are not clearly described. For example, the paper uses a self-prompt approach to extract reference datasets, but it does not specify how many datasets are extracted for each case, which directly relates to the attack costs. This is especially important for commercial LLMs since attacking commercial LLMs will incur high costs in collecting such datasets. Additionally, details on dataset splitting (e.g., how many datasets are used for training the target model) and whether the approach requires training shadow models are missing. If it does not require shadow models, how is the threshold τ in equation (2) determined?

- The paper argues that the proposed probabilistic variation assessment is based on memorization rather than overfitting, but it does not clearly explain the key differences between these two concepts or how the approach improves from the perspective of memorization.

- The reference-based baselines used in the paper are limited and somewhat outdated. Including more advanced baselines, such as [R4], would strengthen the comparison.

- An ablation study to investigate the contributions of the proposed self-prompt approach and probabilistic variation assessment separately would highlight the individual contributions of the paper.


[R1] Carlini, Nicholas, et al. "Membership inference attacks from first principles." 2022 IEEE Symposium on Security and Privacy (2022).

[R2] Bertran, Martin, et al. "Scalable membership inference attacks via quantile regression." Advances in Neural Information Processing Systems (2023).

[R3] Wen, Yuxin, et al. "Canary in a Coalmine: Better Membership Inference with Ensembled Adversarial Queries." International Conference on Learning Representations (2023).

[R4] Shi, Haonan, et al. “Learning-Based Difficulty Calibration for Enhanced Membership Inference Attacks”. IEEE European Symposium on Security and Privacy (2024).

**Questions:**

- Does the proposed approach require training shadow models? If not, how is the threshold τ in equation (2) determined?

- How many datasets are extracted for each case in training the reference models?

- What are the key differences between memorization and overfitting, and how does the proposed approach improve from the perspective of memorization?

**Limitations:**

Please see my above comments.

---

> ### Author Rebuttal · Authors · 2024-08-07
>
> Dear reviewer saYA,
>
> Thank you so much for your thoughtful review and your suggestions for improvement. Below we address the questions raised in your review, the responses to the weaknesses and the reference can be found in subsequent comments:
>
> **Q1,W2: Does the proposed approach require training shadow models? If not, how is the threshold τ in equation (2) determined?**
>
> This is an interesting and meaningful topic, since current MIAs designed for LLMs mostly not discuss how to select an appropriate threshold [1,2,3]. Sharing a similar idea with [4], the proposed method (SPV-MIA) can determine the threshold without a shadow model by picking a quantile on the distribution of confidence scores induced by SPV-MIA on texts that are not used in fine-tuning. In particular, we collected a dataset $\\{\boldsymbol{x}_i\\}_1^n$ from an unrelated task, known to not have been used in fine-tuning. We evaluate the SPV-MIA for each text $\boldsymbol{x}_i$, and record the confidence score $s(\boldsymbol{x}_i)$ that SPV-MIA places it as a member text. We then pick the $1-\alpha$ quantile from the score distribution. Intuitively, a score $s(\boldsymbol{x}_i)$ larger than the $1-\alpha$ quantile indicates that SPV-MIA assigns a confidence on the member text that is higher than a $1-\alpha$ fraction of the texts not used in training. Thus, we can select an expected false positive rate of $\alpha$, then determine the according threshold $\tau$ by measuring the $1-\alpha$ quantile of the score distribution. To further support our statement, we visualize the score distribution of member and non-member texts evaluated by SPV-MIA (see the Figure 2 in PDF file of the global rebuttal).
>
> **Q2,W2: How many datasets are extracted for each case in training the reference models? This is especially important for commercial LLMs.**
>
> As we have summarized in Appendix A.6.4, the number of samples extracted for fine-tuning reference models is set to 10,000 (16 prompt tokens and 112 completion tokens for each). We used the price of GPT-3.5-turbo-0125 as a reference (input: \\$0.50 / 1M tokens, output: \\$1.50 / 1M tokens), and extracting such a dataset from commercial LLMs would cost approximately only \$1.76. Thus, we believe that it is an acceptable value even for attacking commercial LLMs.
>
> Additionally, we have explored the impact of smaller reference datasets (in Section 5.4, Fig. 5(a)), and we quote the relevant results in the table below. This result demonstrates that even in the extracted reference dataset with only 1,000 samples, SPV-MIA achieves performance comparable to 10,000 samples.
>
> Table. The performances of SPV-MIA on LLaMA@AG News while utilizing different scales and sources of extracted reference datasets.
> |Extracted Dataset Scale|1,000|2,000|5,000|10,000|
> |-|-|-|-|-|
> |Irrelevant|0.879|0.882|0.886|0.897|
> |Domain-specific|0.890|0.892|0.895|0.903|
> |Identical-distribution|0.896|0.912|0.915|0.922|
>
> **Q3,W3: Questions aboout memorization.**
>
> > **Q3.1: What are the key differences between memorization and overfitting?**
>
> Although memorization is associated with overfitting, overfitting by itself cannot completely explain some properties of memorization [5,6,7].
> The key differences between memorization and overfitting can be summarized as the following three points:
> * **Occurrence Time**
>  Existing research defines the first epoch when the LLM's perplexity (ppl) on the validation set starts to rise as the occurrence of overfitting [5]. In contrast, memorization begins early [5,6] and persists throughout almost the entire training phase [6,7].
> * **Harm Level**
>  Overfitting is almost universally acknowledged as a detrimental phenomenon in machine learning. However, memorization is not exclusively harmful, and can be crucial for certain types of generalization (e.g., on QA tasks) [8,9].
> * **Avoidance Difficulty**
>  Since memorization occurs much earlier, even if we use early stopping to prevent the model from overfitting, we will still achieve significant memorization [6]. Moreover, since memorization is crucial for certain tasks [8,9], and separately mitigates specific unintended memorization (e.g., verbatim memorization [10]) is a non-trivial task.
>
> > **Q3.2: How does the proposed approach improve from the perspective of memorization?**
>
> Based on the aforementioned discussion, memorization is a more common and robust phenomenon in LLMs compared to overfitting. Therefore, we believe that identifying the memorization footprint left by LLMs on member texts can help improve MIA. Moreover, memorization in generative models [11] or LLMs [2] causes an increased tendency of generative probability density around member records. Thus, we can translate the problem of detecting member texts into identifying texts that are close to the local maximum. Subsequently, we designed a novel metric, probability variation, as an indicator of local maxima and proposed two symmetrical paraphrasing methods to estimate it (Section 4.3). Consequently, our proposed method can enhance the identification of memorization. We have conducted ablation experiments (Appendix A.4, Table 5) to show that our proposed probability variation assessment (PVA) approach can further improve the success rate of MIA compared with neighbour attack. We quote the existing experimental results in the table below.
>
> Table: Results of Ablation Study on LLaMA across three datasets.
> |Dataset|Wiki|AG News|XSum|Avg.|
> |-|-|-|-|-|
> |SPV-MIA (Embedding-based Paraphrasing)|0.956|0.926|0.949|0.944|
> |SPV-MIA (Semantic-based Paraphrasing)|0.951|0.903|0.937|0.930|
> |SPV-MIA (Neighbour Attack)|0.934|0.893|0.928|0.918|
>
> Thanks for your valuable questions for improvement, we will include these discussions in the camera-ready version.

---

> > ### Comment · Reviewer_saYA · 2024-08-12
> >
> > I thank the authors for their detailed response, which has addressed most of my concerns. I do believe the main evaluation should focus on the True Positive Rate (TPR) in the low False Positive Rate (FPR) regime. Please consider including all related experiments in the revised manuscript. I will raise my score accordingly.

---

> > > ### Author Response · Authors · 2024-08-12
> > > **Additional Feedback to Confirm Suggestions Raised by Reviewer saYA**
> > >
> > > Dear reviewer saYA,
> > >
> > > Thank you for your further suggestions. We have organized the TPR@1%FPR and TPR@0.1%FPR corresponding to all baselines and SPV-MIA into tables formatted the same as the current main evaluation (Table 1 in the initial submission). After reviewing the papers you recommended, we agree that TPR in the low FPR should indeed be highlighted as the main evaluation. Therefore, we will follow your advice and replace Table 1 in the initial manuscript with `"Table 2: Evaluation of all baselines and SPV-MIA using TPR@1%FPR"` in the PDF file (see the global comment). Additionally, all related experiments mentioned in this rebuttal will be included in the camera-ready version. We would appreciate it if you could raise the score accordingly.
> > >
> > > Best regards,
> > >
> > > Authors

---

> > > ### Author Response · Authors · 2024-08-14
> > >
> > > Dear reviewer saYA,
> > >
> > > Thank you for providing prompt feedback. We will definitely **highlight and focus** on TPR@Low FPR in the main evaluation and add all related experiments into the revised version.  If you have further suggestions or comments, we would like to address them before the end of the discussion period. Thank you for your time and valuable comments.
> > >
> > > Best regards,
> > >
> > > Authors

---

> ### Author Response · Authors · 2024-08-07
> **(Optional for reviewer to read) The responses to the weaknesses -- Part 1**
>
> To answer the detailed questions of saYA, we provide more information in the form of comments. We understand reviewers are not obligated to read comments. we keep them just in case saYA are interested.
>
> **W1: The main evaluation metrics need to be revised. I suggest using two metrics: the Full Log-scale Receiver Operating Characteristic (ROC) Curve to highlight low false-positive rates, and the TPR at a low FPR.**
>
> We have followed your suggestion and added three new evaluation metrics: the Full Log-scale Receiver Operating Characteristic (ROC) Curve, TPR@1%FPR, and TPR@0.1%FPR. All additional results can be found in the PDF file of the global response. Thank you for your valuable suggestion for improvement. We will update these results to the camera-ready version. We quote partial results in the tables below. Experimental results show that, compared to AUC, SPV-MIA achieves a larger performance margin under new metrics. This indicates that, at lower FPRs, SPV-MIA maintains a higher TPR compared to the baselines.
>
> Table. Evaluation of all baselines and SPV-MIA on LLaMA@AG News Dataset using TPR@1\%FPR and TPR@0.1\%FPR
> |Methods|Loss Attack|Neighbour Attack|DetectGPT|Min-K\%|Min-K\%++|LiRA-Base|LiRA-Candidate|SPV-MIA|
> |------------|-----------|----------------|---------|-------|---------|---------|--------------|-------|
> |TPR@1\%FPR|1.2\%|3.1\%|2.7\%|3.6\%|4.1\%|8.7\%|11.5\%|39.5\%|
> |TPR@0.1\%FPR|0.2\%|0.4\%|0.4\%|0.6\%|1.0\%|1.4\%|1.9\%|23.5\%|
>
> **W4: Including more advanced baselines would strengthen the comparison, such as LDC-MIA [12]**
>
> Thank you for your suggestions regarding the incorporation of more up-to-date baselines. We have carefully reviewed the literature you recommended and believe that LDC-MIA is a related work but not suitable as a new baseline for the following two reasons. However, we will include a thorough discussion of this in the camera-ready version.
> 1. LDC-MIA is designed for classification models and contains modules specifically tailored for classification tasks, and cannot be directly employed for LLMs. For example, the MIA classifier in the LDC-MIA requires the class label of the target sample, which does not exist in text generation tasks. Therefore, adapting LDC-MIA to LLMs is beyond the scope of this work.
> 2. LDC-MIA requires a larger scale auxiliary dataset that has the same distribution as the data used for training the target model. The dataset will be split into three parts: the training and heldout datasets of the shadow model, and the training dataset of the reference model. This is a widely acknowledged unrealistic assumption, especially in the context of LLMs [1,2,3], as their training datasets are typically not publicly available. Our proposed method has avoided this assumption through the self-prompt method. Therefore, it is unfair to compare our algorithm when there is a significant disparity in prior knowledge.
>
>
> However, following your suggestion, we have added two state-of-the-art algorithms (Min-K\%[1] and Min-K\%++[2])  proposed in 2024, specifically designed for LLMs. The supplementary experiments are shown in the PDF file of the global response. We will update these results in the camera-ready version. We quote the partial results in the table below:
>
> Table: Evaluation of representative baselines and SPV-MIA on LLaMA@AG News Dataset using TPR@1\%FPR, TPR@0.1\%FPR and AUC
> |Metric|AUC|TPR@1%FPR|TPR@0.1%FPR|
> |----------------|-----|---------|-----------|
> |Neighbour Attack|0.610|3.1\%|0.4\%|
> |Min-K\%|0.619|3.6\%|0.6\%|
> |Min-K\%++|0.631|4.1\%|1.0\%|
> |LiRA-Base|0.657|8.7\%|1.4\%|
> |LiRA-Candidate|0.714|11.5\%|1.9\%|
> |SPV-MIA|0.903|39.5\%|23.5\%|

---

> ### Author Response · Authors · 2024-08-07
> **(Optional for reviewer to read) The responses to the weaknesses -- Part 2**
>
> To answer the detailed questions of saYA, we provide more information in the form of comments. We understand reviewers are not obligated to read comments. we keep them just in case saYA are interested.
>
> **W5: Conducting an ablation study to investigate the contributions of each proposed component (PVA and PDC) would be beneficial.**
>
> We have shown the ablation study in Appendix A.5.1 for auditing the contributions of the proposed Practical Difficulty Calibration (PDC) and probabilistic variation assessment (PVA) separately. Furthermore, we also have conducted an extensive ablation study in Appendix A.4 to investigate the contributions of our proposed two symmetrical paraphrasing methods compared with neighbour attack, in which we adopt the two symmetrical paraphrasing methods, and neighbour attack as the PVA module, respectively. Thank you for the reminder, we will emphasize the ablation study in the main body of the camera-ready version.
>
> In terms of these ablation studies shown in the Appendix, we quote the experimental results and merge them into one table as follows:
>
> Table: Results of Ablation Study on LLaMA across three datasets.
> |Dataset|Wiki|AG News|XSum|Avg.|
> |--------------------------------------|-----|-------|-----|-----|
> |**SPV-MIA (Embedding-based Paraphrasing)**|**0.956**|**0.926**|**0.949**|**0.944**|
> |SPV-MIA (Semantic-based Paraphrasing)|0.951|0.903|0.937|0.930|
> |SPV-MIA (Neighbour Attack)|0.934|0.893|0.928|0.918|
> |w/o PVA|0.913|0.885|0.919|0.906|
> |w/o PDC|0.653|0.641|0.661|0.652|
>
> The results demonstrate that both PDC and PVA contribute a certain improvement to our proposed method. However, the PDC approach seems to play a more critical role, which can still serve as a valid adversary without the PVA. Thus, in practical scenarios, we can consider removing the PVA to reduce the frequency of accessing public APIs. Additionally, the proposed two paraphrasing methods both yield considerable performance gains compared to the neighbour attack.
>
> # Reference
> [1] Shi, et al. "Detecting Pretraining Data from Large Language Models." ICLR 2024.
>
> [2] Zhang, et al. "Min-K%++: Improved Baseline for Detecting Pre-Training Data from Large Language Models." arXiv 2024.
>
> [3] Mattern, et al. "Membership Inference Attacks against Language Models via Neighbourhood Comparison." ACL 2023.
>
> [4] Bertran, et al. "Scalable membership inference attacks via quantile regression." NeurIPS 2023.
>
> [5] Tirumala, et al. "Memorization without overfitting: Analyzing the training dynamics of large language models." NeurIPS 2022.
>
> [6] Mireshghallah, et al. "An empirical analysis of memorization in fine-tuned autoregressive language models." EMNLP 2022.
>
> [7] Zhang, et al. "Counterfactual memorization in neural language models." NeurIPS 2023.
>
> [8] Tay, et al. "Transformer memory as a differentiable search index." NeurIPS 2022.
>
> [9] Borgeaud, et al. "Improving language models by retrieving from trillions of tokens." ICML 2022.
>
> [10] Ippolito, et al. "Preventing Generation of Verbatim Memorization in Language Models Gives a False Sense of Privacy." INLG 2023.
>
> [11] van den Burg, Williams. "On memorization in probabilistic deep generative models." NeurIPS 2021.
>
> [12] Shi, Haonan, et al. “Learning-Based Difficulty Calibration for Enhanced Membership Inference Attacks”. IEEE European Symposium on Security and Privacy (2024).

---

### Official Review · Reviewer_DLpC · 2024-07-15

**Soundness:** 3
**Presentation:** 3
**Contribution:** 3
**Rating:** 7
**Confidence:** 4

**Summary:**

This paper proposes self-calibrated probabilistic variation (SPV)-MIA, a membership inference attack. The novel ideas that SPV-MIA introduces in the space of LLMs: 1) using paraphrasing to obtain samples around the target sample text in the sample domain, and using paraphrased texts to compute probabilistic variation of the target model around the target, and 2) using self-prompting to generate reference data that SPV-MIA uses to train a reference model which is then used to calibrate the probabilistic variation of the target model on the target samples. Experimental evaluation on three benchmark datasets show that SPV-MIA outperforms existing MIAs.

**Strengths:**

- Multiple interesting ideas that result in a strong, practical MIA on LLMs
- Good and easy to follow MIA

**Weaknesses:**

- Some parts of the paper need support, e.g., how PVA is more general than neighborhood attack.
- Datasets evaluated with are tiny; larger datasets should be used
- Writing can be improved for better comprehension

**Questions:**

- Idea of PVA is similar to neighborhood attack. The paper claims that they generalize the notion of neighborhood attack but end up using the neighborhood attack. This is alright, but how does this generalization help? Is the paper merely depicting neighborhood attack using a set of equations or does the generalization can be somehow helpful to improve the attack success?
- Experiments use small benchmark datasets for fine-tuning which are probably also a part of training data of pre-trained base model; both of these can lead to overestimation of power of proposed MIA. Why do you use small datasets? How does the proposed MIA work for larger fine-tuning datasets?
- About DP results:
    - Can you explain how do you perform DP training?
    - How do the baseline attacks perform for the same DP guarantees?

**Limitations:**

- Writing of the paper is poor; there are too many grammatical mistakes (even in abstract) and many sentences are not properly constructed making it difficult to read the paper.

---

> ### Author Rebuttal · Authors · 2024-08-07
>
> Dear reviewer DLpC,
>
> We deeply appreciate the time you took to review our work and your comments for improvement. Below we address the concerns raised in your review:
>
> **Q1,W1: Questions about the PVA generalization.**
>
> > **Q1.1: The paper claims the generalization of PVA but end up using the neighborhood attack.**
>
>
> The default paraphrasing method used is not the neighbour attack but the proposed semantic-based symmetrical paraphrasing method. Unlike the neighbour attack, each paraphrasing operation of the proposed methods will generate a pair of symmetrical neighbouring samples. In contrast, paraphrasing operations in the neighbour attack produce independent neighbouring samples.
>
> > **Q1.2: How does the generalization of PVA help? Does the generalization can be somehow helpful to improve the attack success?**
>
> The proposed PVA algorithm not only uses a series of generalized equations to reformulate the neighbour attack but also actually improves the attack success. Therefore, the PVA contributes both theoretically and empirically:
>
> * **Empirical contribution** (improve attack success)
>
> We derive two symmetrical paraphrasing methods (Semantic-based and Embedding-based) from PVA. We have conducted an extensive ablation study in Appendix A.4, where the results demonstrate that both two proposed methods achieve considerable performance gains compared with neighbour attack. Thus, the proposed PVA can actually improve the attack performance.
>
> Table: Results of Ablation Study on LLaMA across three datasets.
> |Dataset|Wiki|AG News|XSum|Avg.|
> |-|-|-|-|-|
> |SPV-MIA (Semantic-based Paraphrasing)|0.951|0.903|0.937|0.930|
> |SPV-MIA (Embedding-based Paraphrasing)|0.956|0.926|0.949|0.944|
> |SPV-MIA (Neighbour Comparing)|0.934|0.893|0.928|0.918|
>
> * **Theoretical contribution** (Motivate paraphrasing-based attacks from another perspective)
>
> The existing paraphrasing-based attacks (e.g., Neighbour Attack) are motivated by exploring a substitution of reference models. In this work, we provide another motivation based on model memorization, that is, the member text memorized by the model will tend to be located on the local maximum in the generative distribution [1]. Therefore, this study may inspire the following studies to design more sophisticated methods for inferring membership by characterizing the model memorization. Coincidentally, a recently published work (after the NeurIPS deadline) recommended by reviewer i2LA [2] also proposed an MIA against pre-trained LLMs based on the concept of local maximum detection.
>
> Thank you for the reminder, we will emphasize the contributions of PVA in the main body of the camera-ready version.
>
> **Q2,w2: Questions about small datasets.**
>
> > **Q2.1 Why do you use small datasets? The small datasets maybe a part of pre-training data, which can lead to overestimation of MIA performance.**:
>
> In fact, we employed two datasets (Ag News, Wiki) that are the same as those used in existing studies [3,4], as well as one larger dataset (Xsum). The possibility of overlap between the fine-tuning and pre-training datasets will lead to an underestimation of MIA performance. Since if the fine-tuning dataset is a part of the pre-training data, then both the training and evaluation splits of the fine-tuning dataset will be included in the pre-training data with a large probability. In this scenario, we not only need to detect texts used during the fine-tuning stage but also avoid falsely identifying texts that were only used during the pre-training stage and did not appear in the fine-tuning stage as member texts.
>
> > **Q2.2: How does the proposed MIA work for larger fine-tuning datasets?**:
>
> Although we cannot use larger datasets as the default setting for the overall experiment, we agree that exploring larger fine-tuning datasets would further enhance the quality of our work. Therefore, we used a dataset nearly 10x larger than the current ones, Newsroom, to fine-tune LLaMA as the target model. The results in the table below demonstrate that SPV-MIA still maintains relatively good performance for the larger dataset.
>
> |Metric|SPV-MIA|Neighbour Attack|LIRA-Candidate|
> |-|-|-|-|
> |AUC|0.862|0.545|0.665|
>
>
> **Q3: Can you explain how do you perform DP training? How do the baseline attacks perform for the same DP guarantees?**:
>
> We follow the private-transformers [5], using ghost clipping to perform DP training in a memory-saving manner. We have conducted three representative baseline attacks for the DP-SGD model (LLaMA@Ag News) and compared it with our method (SPV-MIA). The results are provided in the table below. The results demonstrate that SPV-MIA consistently maintains the best MIA performance over different settings of the privacy budget. Thank you for your suggestion to improve our work further. We will update these results to the camera-ready version.
>
> |Privacy Budget $\epsilon$|15|30|60|+inf|
> |-|-|-|-|-|
> |Loss Attack|0.523|0.551|0.568|0.580|
> |Neighbour Attack|0.542|0.564|0.587|0.610|
> |LIRA-Candidate|0.611|0.655|0.684|0.714|
> |SPV-MIA|0.766|0.814|0.852|0.903|
>
> # Reference
> [1] van den Burg, Williams. "On memorization in probabilistic deep generative models." NeurIPS 2021.
>
> [2] Zhang, et al. "Min-K%++: Improved Baseline for Detecting Pre-Training Data from Large Language Models." arXiv 2024.
>
> [3] Mattern, et al. "Membership Inference Attacks against Language Models via Neighbourhood Comparison." ACL 2023.
>
> [4] Mireshghallah, et al. "An empirical analysis of memorization in fine-tuned autoregressive language models." EMNLP 2022.
>
> [5] Li, et al. "Large Language Models Can Be Strong Differentially Private Learners." ICLR 2021.

---

> ### Author Response · Authors · 2024-08-14
>
> Dear Reviewer DLpC,
>
> Thank you for your positive evaluation of our work and for increasing your score from 6 to 7. We appreciate your support and the time you've taken to assess our submission. Your feedback has been invaluable in helping us refine our paper.
>
> Best regards,
> The Authors

---

### Official Review · Reviewer_v2Kc · 2024-07-21

**Soundness:** 3
**Presentation:** 3
**Contribution:** 3
**Rating:** 7
**Confidence:** 4

**Summary:**

This paper studies the membership inference attack on large language models finetuned on private data. Instead of reusing other pre-trained public large language models, the paper proposes a way to generate a reference dataset and finetune this dataset to attain a reference model. With this reference model, the paper further calculates the score by probabilistic variation assessment: symmetrically rephrasing the target data and calculating the average score as the final score. As evaluated in the experiment, the proposed attack has a large margin above the baselines. The paper also conducted ablation studies to understand the design of important and demonstrates the robustness of the algorithm.

**Strengths:**

1. The algorithm is novel. It proposes a way to generate the reference dataset via self-prompting, which leads to a high-quality reference model for a better membership inference attack.
2. The results seem very promising. The performance of the proposed attack has a large margin over the baselines.
3. The algorithm is robust to assuming the domain-specific public data. Even if it is relaxed to irrelevant data, the proposed attack still achieve good performance.

**Weaknesses:**

1. It is not fully in the black-box setting, because it assumes the knowledge of the same pre-trained model and the access of its parameters.
2. The algorithm has two novel parts: propose a way to generate the reference dataset for learning a reference model and a novel score similar to probabilistic variation assessment. It might be important to study how much each component contributes to the final improvements.

**Questions:**

See the weakness.

**Limitations:**

The paper has well-discussed the limitations.

---

> ### Author Rebuttal · Authors · 2024-08-07
>
> Dear reviewer v2Kc,
>
> Thank you so much for your thoughtful review and overall positive comments. Below we address the concerns raised in your review:
>
> **W1: It is not fully in the black-box setting, since it assumes the knowledge of the same pre-trained model and the access of its parameters.**
>
> In fact, SPV-MIA does not require access to the parameters of the target model, indicating that we do not use a white-box setting. However, we are not using a pure black-box setting either, since SPV-MIA requires the following adversary knowledge (two APIs):
> * The access to the query API with response results (generated texts and logits) of the target model.
> * [+] The access to the fine-tuning API of the pre-trained version of the target model.
>
> Here, we only add a fine-tuning API compared with existing works [1,2,3,4]. Overall, we do not require white-box access to the parameters of this pre-trained model; an API that allows the adversary to upload a customized dataset for fine-tuning would also meet the conditions. Therefore, our proposed method is not only applicable to open-source models with publicly available parameters but also for closed-source models that provide fine-tuning APIs (such as ChatGPT [5]). We will include the above clarification in the camera-ready version.
>
> Furthermore, to further allay your concerns, we consider a truly black-box scenario in which the adversary is constrained to fine-tune a pre-trained model different from the target model as the reference model. We employ LLaMA-7B fine-tuned over the Agnew dataset as the target model, and let the adversary fine-tune over different pre-trained models (GPT-J, Falcon, and LLaMA are adopted) to prepare the reference model. The results are shown in the table below. From the experimental results in the table, we can find that the proposed method still maintains a large performance margin over the baselines (LiRA-Base, LiRA-Candidate) without this constraint. We will include these results in the camera-ready version.
>
> |Metric|AUC|TPR@1%FPR|TPR@0.1\%FPR|
> |----------------------|-----|---------|-----------|
> |SPV-MIA (GPT-J)|0.832|19.3\%|7.5\%|
> |SPV-MIA (Falcon)|0.812|21.7\%|8.7\%|
> |SPV-MIA |0.903|39.5\%|23.5\%|
> |LiRA-Base |0.657|8.7\%|1.4\%|
> |LiRA-Candidate |0.714|11.5\%|1.9\%|
>
> **W2: Conducting an ablation study to investigate the contributions of each proposed component would be beneficial.**
>
> In Appendix A.5.1, we have shown the ablation study for evaluating the contributions of the proposed Practical Difficulty Calibration (PDC) and probabilistic variation assessment (PVA) separately. Furthermore, we have also conducted a comprehensive ablation study in Appendix A.4 to investigate the contributions of our proposed two symmetrical paraphrasing methods compared with neighbour attack, in which we adopt the two symmetrical paraphrasing methods, and neighbour attack as the PVA module, respectively. Thank you for the reminder, we will emphasize the ablation study in the main body of the camera-ready version.
>
> In terms of these ablation studies shown in the Appendix, we quote the experimental results and merge them into one table as follows:
>
> Table: Results of Ablation Study on LLaMA across three datasets.
> |Dataset|Wiki|AG News|XSum|Avg.|
> |--------------------------------------|-----|-------|-----|-----|
> |**SPV-MIA (Embedding-based Paraphrasing)**|**0.956**|**0.926**|**0.949**|**0.944**|
> |SPV-MIA (Semantic-based Paraphrasing)|0.951|0.903|0.937|0.930|
> |SPV-MIA (Neighbour Attack)|0.934|0.893|0.928|0.918|
> |w/o PVA|0.913|0.885|0.919|0.906|
> |w/o PDC|0.653|0.641|0.661|0.652|
>
> The results demonstrate that both PDC and PVA contribute a certain improvement to our proposed method. However, the PDC approach seems to play a more critical role, which can still serve as a valid adversary without the PVA. Thus, in practical scenarios, we can consider removing the PVA to reduce the frequency of accessing public APIs. Additionally, the proposed two paraphrasing methods both yield considerable performance gains compared to the neighbour attack.
>
> # Reference
> [1] Shi, Weijia, et al. "Detecting Pretraining Data from Large Language Models." The Twelfth International Conference on Learning Representations (2024).
>
> [2] Duan, Michael, et al. "Do membership inference attacks work on large language models?." Conference on Language Modeling (COLM) (2024).
>
> [3] Mattern, Justus, et al. "Membership Inference Attacks against Language Models via Neighbourhood Comparison." Findings of the Association for Computational Linguistics: ACL 2023. 2023.
>
> [4] Zhang, Jingyang, et al. "Min-K%++: Improved Baseline for Detecting Pre-Training Data from Large Language Models." arXiv preprint arXiv:2404.02936 (2024).
>
> [5] Andrew Peng, Michael Wu, John Allard, Logan Kilpatrick, and Steven Heidel. Gpt-3.5 turbo finetuning and api updates, August 2023.

---

> > ### Comment · Reviewer_v2Kc · 2024-08-13
> > **Official Comment by Reviewer v2Kc**
> >
> > Thank authors for the responses, which have addressed my questions. I will keep my positive rate. I would recommend authors better clarify the setting with the assumptions in the revision by adding the discussions above.

---

> > > ### Author Response · Authors · 2024-08-14
> > >
> > > Dear Reviewer v2Kc,
> > >
> > > Thank you for taking the time to review our responses and for maintaining your positive rating of our manuscript. We appreciate your constructive feedback and will ensure that the revised version of our paper includes clearer explanations of the setting and assumptions, incorporating the discussions we've had.
> > >
> > > Best regards,
> > >
> > > The Authors

---

### Author Rebuttal · Authors · 2024-08-07

We have attached the experimental results added according to the reviewers' requirements in the attached PDF file, which includes the following contents.

1. Table1: Evaluation of all baselines and SPV-MIA using AUC scores.
2. Table 2: Evaluation of all baselines and SPV-MIA using TPR@1%FPR.
3. Table 3: Evaluation of all baselines and SPV-MIA using TPR@0.1%FPR.
4. Figure 1: Full log-scale ROC curves of SPV-MIA and the three representative baselines on LLaMAs fine-tuned over three datasets.
5. Figure 2: The distributions of member and non-member records w.r.t MIA metric score ∆m(x).

---

### Decision · Program_Chairs · 2024-09-25

**Decision:**

Accept (poster)

**Comment:**

This work proposes a novel usage of paraphrasing for improving membership inference attacks by using paraphrasing for generating reference datasets and for improving the membership signal using a style of neighborhood querying.

Overall there is consensus that this paper meets the quality threshold for Neurips. The reviewers agree this paper uses interesting ideas to create a novel attack that has clear and substantial improvements over prior work. However, there are some minor issues in that this work shares some ideas from prior work, could have a clearer evaluation with some more baselines, larger datasets, better metrics, and additional ablations. Some of these were addressed in the rebuttal.